# Limits to scalable evaluation at the frontier: LLM as Judge won't beat twice the data

**Florian E. Dorner**[*,1,2,3], **Vivian Y. Nastl**[1,2,3], **and Moritz Hardt**[1,2]

[1]Max Planck Institute for Intelligent Systems, Tübingen [2]Tübingen AI Center [3]ETH Zürich

## Abstract

High quality annotations are increasingly a bottleneck in the explosively growing machine learning ecosystem. Scalable evaluation methods that avoid costly annotation have therefore become an important research ambition. Many hope to use strong existing models in lieu of costly labels to provide cheap model evaluations. Unfortunately, this method of using models as judges introduces biases, such as self-preferencing, that can distort model comparisons. An emerging family of debiasing tools promises to fix these issues by using a few high quality labels to debias a large number of model judgments. In this paper, we study how far such debiasing methods, in principle, can go. Our main result shows that when the judge is no more accurate than the evaluated model, no debiasing method can decrease the required amount of ground truth labels by more than half. Our result speaks to the severe limitations of the LLM-as-a-judge paradigm at the evaluation frontier where the goal is to assess newly released models that are possibly better than the judge. Through an empirical evaluation, we demonstrate that the sample size savings achievable in practice are even more modest than what our theoretical limit suggests. Along the way, our work provides new observations about debiasing methods for model evaluation, and points out promising avenues for future work.

## 1 Introduction

As large models continue to advance in their capabilities, it is increasingly challenging for human experts to evaluate newly released models. Expert data annotation is not only slow and costly. Traditional benchmarking also struggles to keep up with rapidly changing model capabilities across an expanding range of tasks. Yet, as models become more powerful, there is hope that models themselves could become powerful tools for scaling up evaluation. An intriguing idea is to use a strong existing model to provide judgments about other models. Such a "model-as-judge" could provide labels to classification instances, compare model outputs, and replace human annotators across a variety of tasks.

Already implemented in practice in numerous instances, the model-as-judge paradigm, however, runs into some roadblocks. When used as judges, models exhibit a range of biases that can skew model comparisons and result in misleading model rankings. Recently proposed *debiasing* methods, however, promise a compelling way forward. Using a small number of ground truth labels, these methods can potentially debias a large number of model predictions, thus restoring their utility for benchmarking purposes.

Scalable evaluation is most needed—and most challenging—at the *evaluation frontier*: Newly released models for which we have little intuition as of yet. What makes this case so challenging is that the new model is likely better than the judge in some ways. In this work, we address the question whether debiasing methods together with the model-as-judge paradigm can, in principle, provide an adequate solution to scalable evaluation at the frontier. However, our theory, supported by empirical evaluation, makes a sobering prediction: Whenever the judge model performs worse at its task than the evaluated model, the optimal debiasing method is no better than using twice the ground truth data. This shows that, although there is merit to debiasing, at the evaluation frontier its economic gains are not greater than a factor-two savings in annotation cost.

---

[*]Corresponding author: florian.dorner@tuebingen.mpg.de

## 1.1 OUR CONTRIBUTIONS

In our theoretical work we focus on a standard evaluation setup that encompasses classification, question answering, arena-style comparisons, and safety evaluations. Given an input $x$, a model $m$ produces an output $m(x)$ that receives a binary ground-truth score $s(x, m(x))$. For example, in classification the accuracy score is $s(x, m(x))$ is 1 if the model's output $m(x)$ matches the ground-truth label $y(x)$ and 0 otherwise. A judge model provides a binary *proxy score* $\tilde{s}(x, m(x))$. We imagine that obtaining ground-truth scores is costly and proxy scores are significantly cheaper.

For a fixed model $m$, our goal is to estimate its true score $\mathbb{E}\, s(x, m(x))$, where the expectation is taken over an input drawn from a distribution. However due to *judge bias*, simply using cheap proxy scores $\tilde{s}$ can lead to substantially distorted estimates. To motivate the issue, Figure 1 shows how proxy scores can cause highly misleading rankings of model performance.

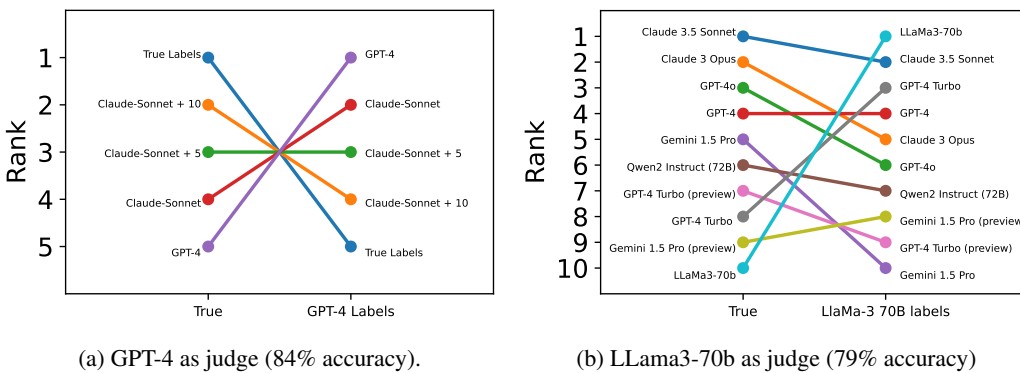

(a) GPT-4 as judge (84% accuracy).      (b) LLama3-70b as judge (79% accuracy)

Figure 1: Model ranks on MMLU based on true labels compared to LLM labels in a semi-synthetic setting (a) and for the top-10 models on HELM by July 2024 (b). Using LLM labels heavily perturbs the ranking despite high judge accuracy.

Rather than solely relying on proxy scores, we assume that we have a small number $n$ of samples of ground-truth scores $s$, in addition to a larger number $N$ of proxy scores $\tilde{s}$. A debiasing method is any estimator that provides an unbiased estimate of the true score from the sample of available proxy scores and ground truth scores. In particular, debiasing methods guarantee that ranking models by their estimated performance produces correct model rankings with high probability.

We focus on sample size savings in terms of the ground-truth scores $s$ that debiasing methods can achieve by making use of the proxy scores $\tilde{s}$.

- We first show that the best possible savings are determined by the Pearson correlation between $s$ and $\tilde{s}$ (Theorem 5).
- Our main result (Theorem 6) then shows that this correlation is small whenever the judge model performs worse than the evaluated model. In that case the best possible outcome is an effective doubling of the number of samples $s$, no matter how many samples of the proxy score $\tilde{s}$ are available (Corollary 7). This result remains true in a slightly weaker sense, when we allow for the proxy $\tilde{s}$ to be continuous rather than binary (Theorem 10).
- In experiments on MMLU and MT-Bench, we empirically confirm that sample size savings of more than a factor two are rare: We only observe them when current state-of-the-art models are used to judge substantially weaker models.

In addition, we show that the commonly reported *agreement rate* between the judge and the ground truth neither meaningfully restricts the judge bias, nor the sample size savings for debiasing methods.

## 2 FORMAL SETUP AND MOTIVATION

In this section, we introduce the formal model we use to analyze the model-as-judge paradigm. We then provide empirical and theoretical evidence that judge bias can indeed cause model-as-judge to produce misleading results, even when judge models are quite accurate.

We focus our analysis on aggregated binary evaluations: Given a prompt $x$, a model $m$ receives a binary score $s(x, m(x))$ depending on its output $m(x)$. As both model outputs and evaluation protocols can be non-deterministic, we treat $s(x, m(x))$ as a random variable, even when the prompt $x$ and model output $m(x)$ are fixed. The model $m$ is then evaluated based on its expected score $\mathbb{E}\, s(X, m(X))$. Here, the expectation is taken over the prompt distribution $X \sim \mathcal{D}$, as well as additional randomness in the scores $s(\cdot, m(\cdot))$ introduced by the evaluation protocol.

This setup covers several important benchmarking and evaluation settings:

- **Accuracy in classification and Q&A benchmarks:** Here, the prompt $x$ is an instance to be classified or a question to be answered. Letting $m(x)$ denote the model's output on input $x$, the accuracy score $s(x, m)$ equals one whenever $m(x)$ equals the label $y(x)$.
- **Arena-style benchmarks:** In this setting $x$ is a prompt. We sample the model's response $m(x)$ and the response $m'(x)$ of another randomly selected model $m'$. The score $s(x, m)$ indicates whether $m(x)$ was judged to be a better response than $m'(x)$.
- **Safety benchmarks:** For this, the prompt $x$ is designed to potentially elicit unsafe behaviour. The score $s(x, m)$ indicates whether the model's response $m(x)$ is safe.

In each of these settings, we can use a language model as a *judge model* to score models more cheaply than via ground truth evaluation. To do so, we compute a *proxy score* $\tilde{s}$ with the help of the judge model. We assume that the proxy score is binary as well. In the classification setting, for example, the judge models could provide the labels for evaluation. The proxy score would then be the classification accuracy with respect to the judge model's labels. In arena comparisons, the LLM could replace human raters. And in safety benchmarks, an LLM could evaluate the safety of model outputs. Although our motivation focuses on language models as the source of proxy scores, our formal work applies to any proxy score, e.g., scores provided by crowdworkers rather than experts.

Precisely modeling the specific relationship between $x$, $s(x, m)$ and $\tilde{s}(x, m)$ is generally infeasible: Even in the simplest case of classification, this would require a precise formal model of not just the judge's prediction $\tilde{m}(x)$, but also the relationship between $x$ and the correct labels $y(x)$. We therefore treat the scoring functions $s$ and $\tilde{s}$ as black boxes, as is standard, and focus on the joint distribution of the two induced random variables $s(m) := s(X, m)$ and $\tilde{s}(m) := \tilde{s}(X, m)$.

As both scores $s(m)$ and $\tilde{s}(m)$ are binary, we only require three parameters to specify their joint distribution:

$$b(m) = \mathbb{P}\{s(m) = 1\}$$
$$p(m) = \mathbb{P}\{\tilde{s}(m) = s(m) \mid s(m) = 1\}$$
$$q(m) = \mathbb{P}\{\tilde{s}(m) = s(m) \mid s(m) = 0\}$$

Here, the parameter $b(m)$ represents the model's expected real score $\mathbb{E}\, s(m)$, while $p(m)$ represents the probability that the evaluations $s(m)$ and $\tilde{s}(m)$ match, conditional on the score $s$ being equal to one (i.e., the *true positive rate*). Similarly, $q(m)$ represents the probability of $s(m) = \tilde{s}(m)$ conditional on $s(m) = 0$ (i.e., the *true negative rate*).

## 2.1 JUDGE BIAS CAN STRONGLY PERTURB MODEL RANKINGS

We argue that estimating the expected proxy score $\mathbb{E}\, \tilde{s}(m)$ instead of the real expected score $\mathbb{E}\, s(m)$ is problematic for benchmarking. In the spirit of Salaudeen & Hardt (2024), a benchmark has two purposes: quantifying model performance and ranking models. In turn, a useful LLM judge should accurately estimate each model's performance, and induce the same ranking of evaluated models as the ground truth. For the first goal, the main quantity of interest is the *judge bias*, defined as

$$\text{JB}(m) := \mathbb{E}(\tilde{s}(m) - s(m)) = (1 - q(m))\,(1 - b(m)) - (1 - p(m))\,b(m)\,.$$

Whenever $\text{JB}(m)$ has large magnitude, the proxy score $\tilde{s}$ misrepresents the performance $b(m)$ of model $m$, even with large sample sizes.

When it comes to ranking models, the relationship between judge bias JB and model rankings is more subtle: If the judge bias $\text{JB}(m)$ was constant across all models, rankings would be unaffected. However, $\text{JB}(m)$ can strongly depend on the evaluated model $m$. If $\text{JB}(m)$ is positive for models

with low performance, and negative for better performing models, using the score of the judge $\tilde{s}$ can easily perturb model rankings. Our next proposition provides such an example inspired by empirical work. Specifically, real-world classifiers are not only highly similar to each other (Mania et al., 2019), Mania & Sra (2020) also observe that when one classifier is better than another on real data, it tends to be strictly better in a *point-wise* sense, for the most part. We therefore consider using a classifier $m$ to evaluate a set $\mathcal{M}$ of strictly better classifiers. In that case, using the proxy evaluations $\tilde{s}$ based on $m$ fully reverses the correct model ranking:

**Proposition 1.** *Consider a binary classifier $\tilde{m}$ and a set of strictly better binary classifiers $\mathcal{M}$ such that $\tilde{m}(x) = y(x)$ implies $m_i(x) = y(x)$ for all $m_i \in \mathcal{M}$. Let $\mathbb{E}\, s(m_i)$ represent the accuracy of model $m_i$ evaluated on the correct labels, and $\mathbb{E}\, \tilde{s}(m_i)$ its accuracy evaluated on predictions of model $\tilde{m}$. Then for $m_i, m_j \in \mathcal{M}$, $\mathbb{E}\, s(m_i) > \mathbb{E}\, s(m_j)$ implies $\mathbb{E}\, \tilde{s}(m_i) < \mathbb{E}\, \tilde{s}(m_j)$.*

We prove Proposition 1 in Appendix C.5. An illustration is provided in Figure 1a. We use GPT-4 labels to evaluate Claude Sonnet, as well as a set of (ficticious) models on MMLU (Hendrycks et al., 2021). The ficticious models are denoted as "Claude-Sonnet + $x$" and designed to be strictly better than Claude. This is done by changing wrong predictions of Claude into correct ones until accuracy is improved by $x\%$. Figure 1b shows a more realistic setting, with LLama-3 70B judging the top-10 MMLU models according to HELM (Liang et al., 2023). While the ranking is not fully reversed, it is strongly perturbed. This effect can also be observed in prior work: For MT-Bench, the ranking of the best models is not consistently preserved when using LLM judges (Zheng et al., 2024).

## 2.2 High agreement rate is insufficient for evaluation

The results from the previous subsection might appear surprising, considering the strong performance of the judge models: Lama-3 70B has $79\%$ accuracy on MMLU, while GPT-4 has $84\%$, and the models used in MT-Bench have $85\%$ agreement with expert annotators. Our next proposition explains how it is possible for model rankings to be strongly perturbed despite accurate judges. Before stating it, we need to formally define the agreement $\mathrm{AG}(m)$ between score $s$ and proxy score $\tilde{s}$ as

$$\mathrm{AG}(m) := \mathbb{P}\{s(m) = \tilde{s}(m)\} = b(m)p(m) + (1 - b(m))\, q(m).$$

In the classification setting, the judge model's accuracy lower bounds the agreement $\mathrm{AG}(m)$: Clearly $s(x, m) = \tilde{s}(x, m)$ whenever the judge model $\tilde{m}$ is correct. But when there are more than two classes, we also get $s(x, m) = \tilde{s}(x, m) = 0$ whenever the true label $y(x)$ and the predictions $m(x)$ and $\tilde{m}(x)$ are all different from each other. In particular, this highlights that the agreement $\mathrm{AG}(m)$ can indeed depend on the evaluated model $m$, rather than just the judge $\tilde{m}$. For binary classification, however, the agreement rate equals the judge model's accuracy and is thus constant across evaluated models. Our next proposition now relates judge bias to agreement:

**Proposition 2.** *Fix any model score $b(m)$. Then for any $r$ with $b(m) \le r$, there are values of $q(m)$ and $p(m)$, such that $\mathrm{AG}(m) = r$ and we obtain positive judge bias $\mathrm{JB}(m) = 1 - r$. Similarly, for any $r$ with $1 - b(m) \le r$, there are values of $q(m)$ and $p(m)$, such that $\mathrm{AG}(m) = r$ and we obtain negative judge bias $\mathrm{JB}(m) = r - 1$.*

Proposition 2 shows that for agreement $\mathrm{AG}(m) = r$, there can be a judge bias of $1 - r$ in either direction. This means that even if $\mathrm{AG}(m) = r$ was constant across models $m$, we could only reliably rank models for which the true score $\mathbb{E}\, s(m)$ differs by more than $2(1 - r)$. As it is common for state-of-the-art models to differ in performance by only low single digit percentages, without further assumptions an agreement rate above $99\%$ (for each evaluated model!) would be required to ensure (asymptotically) correct rankings. For a proof, see Appendix C.6.

## 3 Debiasing LLM judgments

In the previous section, we demonstrated how biased LLM judges can lead to strongly perturbed model rankings. This motivates the need for debiasing methods. We begin this section by discussing a simple sufficient condition to guarantee asymptotically correct rankings: Ensure that our estimates of $\mathbb{E}\, s(m)$ are unbiased for all models $m$. Statistical bias correction methods can make use of a large number of proxy scores and a smaller number of gold standard labels to construct an unbiased estimator. We discuss one such method in detail, and show that it is essentially optimal in terms of estimator variance. Afterwards we prove our main result, i.e., that LLM judges offer limited benefits for evaluating state-of-the-art models.

### 3.1 UNIFORMLY CONTROLLING JUDGE BIAS ENSURES CORRECT RANKINGS

We begin with a simple and natural sufficient condition for correct rankings. This is that all model evaluations are approximately unbiased. The next proposition shows that this, in fact, entails correct rankings in expectation and hence also with a sufficient amount of data.

**Proposition 3.** *Let $M$ be a finite set of models such that for any $m, m' \in \mathcal{M}$*

$$| \mathbb{E}(s(m)) - \mathbb{E}(s(m'))| \geq \epsilon.$$

*Let $\hat{\theta}$ be an estimator for $\mathbb{E}\, s(m)$ such that $\mathrm{Var}\, \hat{\theta}(m)$ converges to zero in the dataset size. Then if $| \mathbb{E}\, \hat{\theta}(m) - \mathbb{E}(s(m))| < \frac{\epsilon}{2}$ for all models $m$, using $\hat{\theta}$ to rank models yields the correct ranking with high probability.*

*Proof.* By construction, $\mathbb{E}\, \hat{\theta}(m)$ induces the same model rankings as $\mathbb{E}\, s(m)$. Chebyshev's inequality and a union bound imply correct rankings as $\mathrm{Var}\, \hat{\theta}(m)$ goes to zero. □

Proposition 3 implies that if we could guarantee sufficiently small estimator bias, a large amount of proxy samples $\tilde{s}$ would yield correct rankings with high probability. However, it is important for bias to be small *for all evaluated models.* In light of the worst-case results from Proposition 2, we cannot rely on this to be true when solely basing our estimates on the proxy scores $\tilde{s}$.

### 3.2 BACKGROUND ON PREDICTION POWERED INFERENCE

We next discuss how to leverage a small amount of ground truth samples $s$ to debias the proxy scores $\tilde{s}$. This yields an unbiased estimator $\hat{\theta}$ for $\mathbb{E}\, s$, and thus asymptotically correct model rankings.

Specifically, we follow the Prediction Powered Inference (PPI) framework (Angelopoulos et al., 2023a) and an for our case equivalent method by Chaganty et al. (2018): Alongside a large iid sample of model judgments $\tilde{s}_i(m) = \tilde{s}(x_i, m)$ for integer $i \in [1, N+n]$, we assume access to the corresponding ground truth label $s_j(m) = s(x_j, m)$ for a small subset of integer $j \in [1, n]$. With this, we can construct an unbiased estimator $\hat{\theta}^{PP}$ for $\mathbb{E}\, s(m)$ by using the small parallel sample to estimate $\mathrm{JB}(m)$, and subtracting it from our estimate of $\mathbb{E}\, \tilde{s}(m)$. The PPI estimator thus equals

$$\hat{\theta}^{PP}(x) = \frac{1}{N}\sum_{i=n+1}^{N+n}\tilde{s}_i(m) + \frac{1}{n}\sum_{j=1}^{n}\left(s_j(m) - \tilde{s}_j(m)\right).$$

Clearly, the PPI estimator is unbiased

$$\mathbb{E}\, \hat{\theta}^{PP} = \mathbb{E}\, \tilde{s}(m) + \mathbb{E}\, s(m) - \mathbb{E}\, \tilde{s}(m) = \mathbb{E}\, s(m),$$

and has variance

$$\mathrm{Var}\, \hat{\theta}^{PP} = \frac{1}{N}\mathrm{Var}\, \tilde{s}(m) + \frac{1}{n}\mathrm{Var}(\tilde{s}(m) - s(m)).$$

Whenever $N \gg n$, the first term on the right hand side is small. Moreover, when $s(m)$ and $\tilde{s}(m)$ are strongly correlated, the second term is small. Assuming both, we have $\mathrm{Var}\, \hat{\theta}^{PP} < \mathrm{Var}\, \hat{\theta}^{GT}$, where $\hat{\theta}^{GT}$ is the ground-truth sample average estimator $\hat{\theta}^{GT} = \frac{1}{n}\sum_{j=1}^{n} s_j(m)$.

However if the correlation between $s(m)$ and $\tilde{s}(m)$ is small, the inequality is reversed. This issue can be fixed by interpolating between the PPI estimator and the classical estimator (Chaganty et al., 2018; Angelopoulos et al., 2023b), setting

$$\hat{\theta}^{PP}_\lambda = \lambda\hat{\theta}^{PP} + (1-\lambda)\hat{\theta}^{GT}.$$

As a linear combination of unbiased estimators this remains unbiased, but we can now optimize over $\lambda$ to minimize estimator variance. We obtain the optimum value $\lambda^* = \mathrm{Cov}(s(m), \tilde{s}(m)) / \left(\left(1+\frac{n}{N}\right)\mathrm{Var}\, \tilde{s}(m)\right)$. At this $\lambda$, the estimator $\hat{\theta}^{PP}_{\lambda^*}$ never increases variance compared to the classical ground truth estimator:

$$\mathrm{Var}\, \hat{\theta}^{PP}_{\lambda^*} = \frac{1}{n}\mathrm{Var}\, s(m) - \frac{1}{n+\frac{n^2}{N}}\frac{\mathrm{Cov}(s(m), \tilde{s}(m))^2}{\mathrm{Var}(\tilde{s}(m))} \leq \frac{1}{n}\mathrm{Var}\, s(m) = \mathrm{Var}\, \hat{\theta}^{GT}.$$

In the following subsection, we discuss to what extent the PPI estimator $\hat{\theta}^{PP}_{\lambda^*}$ can reduce variance or equivalently improve sample efficiency, compared to the ground truth estimator $\hat{\theta}^{GT}$.

### 3.3 THE SAMPLE EFFICIENCY FACTOR $\tau$

As the variance of $\hat{\theta}^{GT}$ scales as $\Theta(1/n)$, using an estimator $\hat{\theta}$ with $r = \frac{\operatorname{Var} \hat{\theta}}{\operatorname{Var} \hat{\theta}^{GT}}$ is equivalent to increasing the ground-truth sample size by a factor of

$$\tau(\hat{\theta}) := \frac{\operatorname{Var} \hat{\theta}^{GT}}{\operatorname{Var} \hat{\theta}} = \frac{1}{r}.$$

We call $\tau(\hat{\theta})$ the *sample efficiency factor* of the estimator $\hat{\theta}$. It is our main quantity of interest. In order to ensure it is well-defined, we assume $b(m), p(m), q(m) \in (0, 1)$ for the remainder of the text. Our next proposition provides an upper bound on the sample efficiency factor $\tau(\hat{\theta}_{\lambda^*}^{PP})$ of PPI, based on the squared Pearson correlation between $s$ and $\tilde{s}$:

**Proposition 4.** *The sample efficiency factor for the PPI estimator is upper bounded by*

$$\tau(\hat{\theta}_{\lambda^*}^{PP}) \le \frac{1}{1 - \rho(s(m), \tilde{s}(m))^2},$$

*where*

$$\rho(s(m), \tilde{s}(m))^2 = \frac{b(m)}{(1 - b(m))} \frac{(p(m) - \mathbb{E}\, \tilde{s}(m))^2}{\mathbb{E}\, \tilde{s}(m)(1 - \mathbb{E}\, \tilde{s}(m))}.$$

This upper bound is large whenever $\rho^2$ is large. However, the sample efficiency factor $\tau(\hat{\theta}_{\lambda^*}^{PP})$ is finite unless we have perfect correlation $\rho^2 = 1$. For any correlation bounded away from 1, the proxy samples can only provide a constant factor improvement. Scaling them up without also scaling the number of ground truth samples has limited benefits. The proof can be found in Appendix C.7.

### 3.4 PPI IS NEAR-OPTIMAL FOR BINARY EVALUATIONS

The limited sample efficiency gains of PPI compared to the classical estimator raise the question whether we can find an unbiased estimator with lower variance. For black-box estimators that do not model the prompt-conditional distribution $\mathbb{P}[(s(m), \tilde{s}(m))|x]$, we show that the answer is *no*:

**Theorem 5.** *Let $\Theta$ be the set of all unbiased estimators for $\mathbb{E}\, s(m)$ that observe $n$ joint samples $(s(m), \tilde{s}(m))$ and $N$ independent proxy samples $\tilde{s}(m)$. Then, any $\hat{\theta} \in \Theta$ fulfills the variance bound*

$$\operatorname{Var} \hat{\theta} \ge \operatorname{Var} \hat{\theta}_{\lambda^*}^{PP}.$$

*This means that the sample efficiency factor of $\hat{\theta}$ is bounded by*

$$\tau(\hat{\theta}) \le \max_{\hat{\theta} \in \Theta} \tau(\hat{\theta}) = \tau(\hat{\theta}_{\lambda^*}^{PP}) \le \frac{1}{1 - \rho(s(m), \tilde{s}(m))^2}.$$

Theorem 5 follows from an application of the Cramér-Rao bound, combined with extensive algebraic manipulation that was assisted by a computer algebra system. Appendix C.1 has the details.

### 3.5 LIMITED GAINS FOR EVALUATING STATE-OF-THE-ART MODELS

With Theorem 5 at hand, we can analyze the best sample efficiency factor $\tau$ an unbiased estimator can achieve by making use of model judgments $\tilde{s}$. In this subsection, we focus on the evaluation of frontier models that outperform older models, including the models used to judge them. We use the evaluated model's score $b(m)$ and the judge's agreement rate $\mathrm{AG}(m)$ as indicators for the respective model's capabilities. We focus on tasks for which performing and judging the task are of similar difficulty, such that these indicators are comparable. As a prime example, for binary classification, $\mathrm{AG}(m)$ equals the judge's accuracy, while $b(m)$ equals the evaluated model's accuracy.

We thus capture the evaluation of frontier models with the assumption that the evaluated model's score $b(m)$ is larger than the judge's agreement rate $\mathrm{AG}(m)$. The next theorem shows that the squared correlation $\rho(s(m), \tilde{s}(m))^2$ is at most one half in this setting.

**Theorem 6.** *Assume $0.5 \le \mathrm{AG}(m) \le b(m)$. Then,*

$$\rho(s(m), \tilde{s}(m))^2 \le 0.5.$$

We prove Theorem 6 in Appendix C.2. Combining Theorem 6 with Theorem 5 allows us to bound the sample efficiency gains that *any* unbiased estimator can achieve for evaluating frontier models:

**Corollary 7.** *Assume* $0.5 \leq \mathrm{AG}(m) \leq b(m)$. *Then,*

$$\tau_{\max} = \max_{\hat{\theta} \in \Theta} \tau(\hat{\theta}) \leq 2.$$

Corollary 7 implies that when evaluating state-of-the-art models, the best we can expect from using LLM judges is a factor-two improvement in sample efficiency. This is unless the judge's task is substantially easier than the evaluated model's task.

### 3.6 High agreement rate is insufficient for sample efficiency gains

Theorem 6 holds for arbitrarily high levels of agreement. Therefore, the theorem shows that high agreement rates are not just insufficient for avoiding judge bias—they're also insufficient for obtaining a meaningful sample efficiency factor $\tau$. Our next proposition goes a step further and shows that the agreement $\mathrm{AG}(m)$ does not provide *any* lower bound on the sample efficiency factor $\tau$, even when the judge bias $\mathrm{JB}(m)$ is zero.

**Proposition 8.** *For any agreement rate* $0.5 \leq \mathrm{AG}(m) < 1$*, there exist values of* $b(m), p(m), q(m) \in (0, 1)$ *such that*

$$\mathrm{JB}(m) = \rho(s(m), \tilde{s}(m))^2 = 0 \text{ and thus } \tau_{\max} = 1.$$

We prove the proposition in Appendix C.8. It implies that in some cases despite access to $\tilde{s}$, no unbiased estimator has less variance than $\hat{\theta}^{GT}$. It might be surprising that there are cases in which we cannot make productive use of the proxy evaluations $\tilde{s}$ despite zero judge bias. However, without further assumptions, $\mathrm{JB}(m)$ can vary wildly between models, so that we cannot skip estimating it anew for each model. Now, even if $\tilde{s}$ has zero judge bias, the estimator does not "know" that, but rather has to estimate $\mathrm{JB}(m)$ from joint observations $(s, \tilde{s})$. In the configuration from the proposition, estimating the judge bias $\mathrm{JB}(m)$ turns out as hard as directly estimating the real score $\mathbb{E}\, s(m)$.

### 3.7 Experiments

So far, we have shown that even optimal debiasing yields limited sample efficiency gains at the frontier (Section 3). In this section, we have a more detailed look into the sample efficiency gains on popular benchmarks that can be achieved by using current flagship models as judges.

Our experiments in this section focus on two settings: Multiple-choice question answering on MMLU (Hendrycks et al., 2021) and chatbot evaluation on MT-Bench (Zheng et al., 2024). Additional experiments on TruthfulQA can be found in Appendix B.2. In all cases, we calculate the empirical correlation $\rho(s, \tilde{s})$ between the score $s$ and the proxy $\tilde{s}$ to obtain and plot an upper bound for the sample efficiency factor $\tau_{\max}$ based on Theorem 5.

We first focus on MMLU, a multiple-choice question answering benchmark for the world knowledge of language models. We obtain model predictions from the HELM (Liang et al., 2023) leaderboard. We then aggregate all subtasks of MMLU into a single test set and set $s$ to equal the accuracy score $s(x, m) = \mathbb{I}(m(x) = y(x))$. Here $\mathbb{I}$ is the indicator function, $m(x)$ is the model's answer and $y(x)$ is the correct answer. For model-based evaluations using $\tilde{m}$ as a judge, we replace the correct answer $y(x)$ by the judge's prediction $\tilde{m}(x)$, setting $\tilde{s}(x, m) = \mathbb{I}(m(x) = m'(x))$.

In the second setting, we evaluate model performance on MT-Bench, an arena-style benchmark designed to evaluate the ability of models to follow instructions. The benchmark comes with results for six models. For each pair of these models $m$ and $m'$, both models are queried with the same prompt $x$. The test set for each model $m$ then consists of all triples of the form $(m, m', x)$, for other models in the benchmark $m'$ and prompts $x$. For each triple in the test set, an expert evaluator decides, whether model $m$ produced a better response than $m'$. In that case, the score $s(x, m)$ for model $m$ is one, otherwise it is zero. For the proxy score $\tilde{s}(x, m)$ we replace the expert's answer by the GPT-4 response provided by Zheng et al. (2024). For more details, consider Appendix B.

Figures 2 and 3 show the optimal sample efficiency factor $\tau_{\max}$ on MMLU and MT-Bench respectively. The sample efficiency factor $\tau_{\max}$ consistently stays below the value of two suggested by

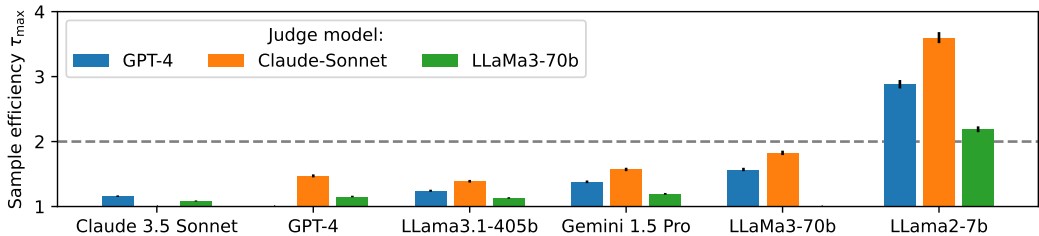

Figure 2: Best possible sample efficiency factor $\tau_{\max}$ according to Theorem 5, using different judges (colors) evaluating different models (x-ticks) on MMLU. Error bars show $90\%$ confidence intervals. Sample efficiency gains stay below two, unless SOTA models are used to evaluate weak models.

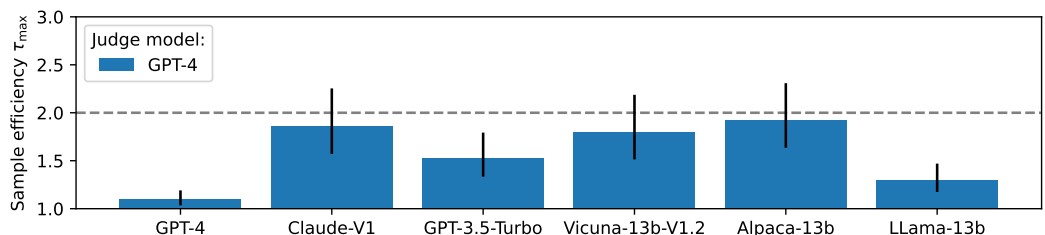

Figure 3: Best possible sample efficiency factor $\tau_{\max}$ according to Theorem 5, using GPT-4-as-a-Judge, evaluating different models on MT-Bench. Error bars show $90\%$ confidence intervals. Sample efficiency get close to two in some cases, but consistently stay below that value.

Corollary 7, except when current flagship models are used to judge the significantly worse LLama2-7b on MMLU. This exception is not surprising, as in that case $\text{AG} \gg b$ such that the assumptions for Corollary 7 are heavily violated. Interestingly, $\tau_{\max}$ stays below two in all other cases, even when stronger models like GPT-4 are used to evaluate weaker models like LLama3-70B. This suggests that the upper bound on $\tau_{\max}$ from Corollary 7 is fairly robust. We also note that $\tau_{\max}$ is often substantially smaller than two, especially on MMLU. This highlights that Corollary 7 provides a best-case upper bound rather than a guarantee that $\tau_{\max}$ will be close to two in practice.

## 4 GOING BEYOND BINARY EVALUATIONS.

So far we have assumed the proxy score $\tilde{s}$ to be binary. In this section, we relax this assumption. By not forcing the judge to fully commit to a single answer, we hope to obtain a more useful proxy $\tilde{s}$. For example, in a Q&A task, we can make use of the uncertainty of a judge model $m'$ by setting $\tilde{s}(x, m) = \mathbb{P}_{m'(x)}(m(x))$ equal to the probability the judge $m'$ assigned to the model answer $m(x)$.

In terms of analysis, the non-parametric joint likelihood of $(s, \tilde{s})$ resulting from non-binary proxies makes it exceedingly hard to prove an analogon to Theorem 5 for all unbiased estimators. Instead, we focus on the PPI estimator, $\hat{\theta}_{\lambda^*}^{PP}$. It is a natural choice of estimator, given its proven optimality in the special case of binary proxies $\tilde{s}$. In order to state a condition similar to $\text{AG} \leq b$ from Theorem 6, we assume the proxy $\tilde{s}$ to be bounded in $[0, 1]$ and define the soft agreement as

$$\text{SO}(\tilde{s}) \coloneqq \mathbb{E}\, s\tilde{s} + (1-s)(1-\tilde{s}).$$

For binary scores $\tilde{s}$, the soft agreement $\text{SO}(\tilde{s})$ simply reduces to the agreement $\text{AG}(\tilde{s})$. For non-binary proxy scores $\tilde{s}$, we interpret them as probability estimates, such that SO equals the estimated probability of the real score $s$. To make this interpretation valid, we focus on the recalibrated proxy $R(\tilde{s}) = \mathbb{P}(s = 1|\tilde{s})$. The following proposition suggests a way to generalize the condition $\text{AG} \leq b$:

**Proposition 9.** *For any binary proxy $\tilde{s}$ such that $0.5 \leq \text{AG} \leq b$, we have $0.5 \leq \text{SO}(R(\tilde{s})) \leq b$.*

Proposition 9 states that if the agreement $\text{AG}(\tilde{s})$ of a binary proxy $\tilde{s}$ is below the evaluated model's accuracy $b$, the same is true for the soft accuracy $\text{SO}(R(\tilde{s}))$ of the recalibrated proxy $R(\tilde{s})$. Based on this, we use $\text{SO}(R(\tilde{s})) \leq b$ to generalize the condition $\text{AG}(\tilde{s}) \leq b$ from Theorem 6, and prove:

**Theorem 10.** *For any proxy score $\tilde{s}$ with $\mathrm{SO}(R(\tilde{s})) \leq b$, we have $\rho^2(s, \tilde{s}) \leq 0.5$. Correspondingly, the sample efficiency of PPI is bounded: $\tau(\hat{\theta}_{\lambda^*}^{PP}) \leq 2$.*

We prove more general versions of Proposition 9 and Theorem 10 in Appendix C.3. In these, we allow for the judge to make $\epsilon$ times as many mistakes as the evaluated model for $\epsilon \leq 1$ rather than being strictly worse. In that case, we obtain a maximal sample efficiency factor $\tau(\hat{\theta}^{PP})$ of $\frac{2}{\epsilon}$.

**Experiments on MMLU**   Using the prompt format from HELM, we extract LLama3.1-405B's next-token predictions $\tilde{p}_x$ for all questions $x$ in MMLU. From these, we take the probabilities $\tilde{p}_x(t)$ corresponding to the four tokens $t \in \{A, B, C, D\} =: T$ representing the answers to question $x$. We then renormalize them to $\mathbb{P}_{m'(x)}(t) = (\tilde{p}_x(t))/(\sum_{t' \in T} \tilde{p}_x(t'))$ and define the proxy score $\tilde{s}(x, m)$ as the probability $\mathbb{P}_{m'(x)}(m(x))$ assigned to the evaluated model's answer $m(x)$, as described above.

Figure 4 shows the sample efficiency factor $\tau(\hat{\theta}_{\lambda^*}^{PP})$ for PPI in the limit of infinite unlabeled data $N \rightarrow \infty$. As expected, using the non-binary proxy consistently improves sample efficiency. However, as suggested by Theorem 10, the sample efficiency factor $\tau(\hat{\theta}_{\lambda^*}^{PP})$ remains below two when we use LLama3.1-405B to evaluate the stronger Claude 3.5. As in the previous experiments, the same is true for evaluating slightly weaker models such as GPT-4 or Gemini 1.5. This shows that our main claim of "no sample efficiency gains of more than a factor two at the frontier" is robust. That said, unlike in the binary case, the factor two is already (slightly) exceeded for LLaMa3-70B (which is 5% less accurate than LLama3.1-405B), rather than just for the much weaker LLaMa2-7B.

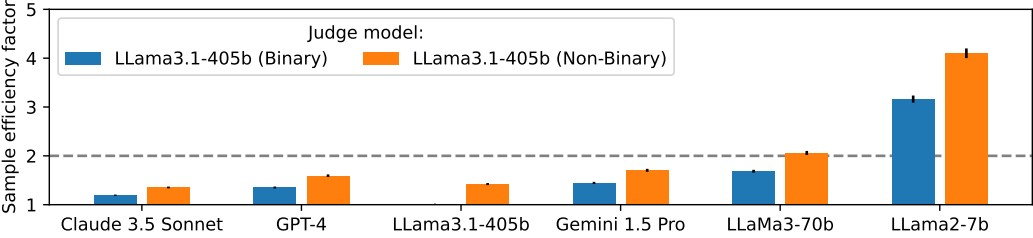

Figure 4: Sample efficiency factor $\tau(\hat{\theta}_{\lambda^*}^{PP})$ for PPI in the $N \rightarrow \infty$ limit, using LLama3.1-405B-as-Judge on MMLU, with binary and non-binary scores. Error bars: 90% confidence intervals. Non-binary scores improve the sample efficiency factor $\tau(\hat{\theta}_{\lambda^*}^{PP})$, but it stays below two at the frontier.

## 5   RELATED WORK

**LLM-as-a-Judge.**   With the success of large transformer-based language models, the idea of using model predictions to provide feedback for another model has become very popular. Initially, specialized fine-tuned models were used to provide training signals for a second model (Ouyang et al., 2022; Bai et al., 2022; Dorner et al., 2023). The approach was quickly expanded to using general purpose LLMs like GPT-4 (Achiam et al., 2023), not just for providing training signal but also for evaluating other models. This new paradigm, dubbed LLM-as-a-Judge, has been applied to evaluate a variety of model capabilities (Yu et al., 2023; Chiang & Lee, 2023; Fu et al., 2023; Li et al., 2024; Weyssow et al., 2024; Raju et al., 2024; Vu et al., 2024; Kumar et al., 2024). In some cases, not just the ratings, but also the prompts given to evaluated models are designed by LLMs (Bai et al., 2024). LLM-as-a-Judge is often paired with another emerging paradigm for model evaluation: As generative tasks often do not have a single correct answer, models are evaluated in so-called arena benchmarks (Chiang et al., 2024), where different models' responses to the same prompts are ranked to determine the best model. Beyond evaluating models, LLM judges have also been employed to evaluate red-teaming (Mazeika et al., 2024) and jailbreaks (Souly et al., 2024). The use of LLM judges rather than experts or crowd workers is often justified by high agreement rates (Gilardi et al., 2023; Zheng et al., 2024) between both types of judges. However, Thakur et al. (2024) find that seemingly high agreement does not necessarily imply accurate judge scores. In this work, we provide a theoretical justification for that finding and show that agreement is not necessarily a good indicator of judge quality, even when judge scores are debiased via PPI.

**Bias in LLM judges.** LLM judges can be biased in numerous ways, making their evaluations unreliable: Their outputs often correlate poorly with expert annotations (Bavaresco et al., 2024; Koo et al., 2023). Models are known to rate their own outputs more favorable than the outputs of other models (Liu et al., 2023; Panickssery et al., 2024), prefer longer outputs and outputs containing lists, regardless of quality (Dubois et al., 2024b; Wei et al., 2024). They also exhibit choice-order bias (Dominguez-Olmedo et al., 2023; Wang et al., 2023; Shi et al., 2024) as well as a variety of other biases (Koo et al., 2023). While correcting for these known biases (Dubois et al., 2024a; Zheng et al., 2024) is an important first step, manually enumerating and correcting for all judge biases appears infeasible. As one potential alternative, Jung et al. (2024) propose to combat judge bias by having judges abstain based on their confidence. However, if abstention correlates with the evaluated model's performance, this approach can introduce its own biases to model evaluations.

**Debiasing methods.** Usually, the biases listed above are found by identifying patterns in how LLM judgments deviate from (a smaller set of) ground truth labels. Instead of trying to identify and fix specific biases, another line of work uses ground truth labels to directly estimate the bias an LLM judge introduces and correct for it. This approach was already suggested by Chaganty et al. (2018) for debiasing classic automated NLP metrics like BLEU (Papineni et al., 2002) and ROUGE (Lin & Och, 2004). The authors find that their method, which is essentially equivalent to PPI, only improved data efficiency by around 10% using 2018's automated metrics. In addition, they show that their method has optimal worst-case variance. In comparison to their worst-case result, our Theorem 5 shows that $\hat{\theta}_{\lambda^*}^{PP}$ is *always* optimal in our binary evaluation setting.

For modern LLM-based metrics, using PPI has been suggested by Boyeau et al. (2024) and Chatzi et al. (2024), and applied as part of more complicated evaluation pipelines (Saad-Falcon et al., 2023; Tyser et al., 2024). Furthermore, Fisch et al. (2024) combine PPI with stratified sampling for model evaluation. These works consistently show that PPI improves efficiency in terms of ground truth labels, but gains in effective sample size rarely exceed 50% and are always below 100%. Our upper bounds on the sample efficiency factor $\tau_{\max}$ show that gains larger than that are indeed unlikely, especially when evaluating state-of-the-art models.

## 6 DISCUSSION

Our results show that for evaluating frontier models, LLM judges might fall short of the promise of largely replacing expert labelers: While doubling the effective sample size can be useful for practitioners, the order of magnitude of required ground truth labels remains the same with and without access to LLM judges. That said, there are ways to circumvent our negative results:

First, our results pertain to uniform sampling rather than more sophisticated sampling strategies. Approaches like stratified PPI (Fisch et al., 2024) might be able to obtain a somewhat better sample efficiency factor, for example by breaking the assumptions of Theorem 6 per-stratum. That said, Theorem 5 still applies per-stratum. Empirical results presented in Appendix B.3 suggest that at the frontier, stratified PPI rarely improves sample efficiency by more than a factor of two, compared to standard stratified sampling. Second, Theorem 6 only holds if the proxy $\tilde{s}$ is less capable at predicting $s$ than the evaluated model $m$ is at its task. Thus, for tasks in which evaluation is substantially easier than the task itself, or tasks for which data to train a specialized evaluator is abundant, gains in sample efficiency of more than a factor two are possible even when evaluating models at the frontier. In addition, weaker models might find their way into many applications due to cost saving reasons, such that efficient evaluations of these models are still valuable.

Finally, we would like to reiterate that our results do not only apply to LLM judges, but any form of biased evaluators including (poorly instructed) crowdworkers. Whenever the crowdworker majority does not consistently agree with the desired ground truth labels, bias correction is required for valid evaluation results. Similarly, if the crowdworker majority vote is less accurate than the evaluated model, Theorem 6 implies that crowdworker labels are of low value compared to ground truth labels.

## 7 ACKNOWLEDGEMENTS

We would like to thank Anastasios Angelopoulos, Amin Charusaie, Yatong Chen, André Cruz and Guanhua Zhang for helpful discussions and/or feedback on draft versions of this work. Florian

Dorner and Vivian Nastl are grateful for financial support from the Max Planck ETH Center for Learning Systems (CLS).

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

## A    ADDITIONAL THEORETICAL RESULTS

### A.1    SAMPLE EFFICIENCY FACTOR FOR STRICTLY BETTER CLASSIFIERS

In the case of strictly better classifiers discussed in Proposition 1, we can provide an even stronger upper bound: When using a classifier $\tilde{m}$ with accuracy $x > 0.5$ to evaluate a strictly better classifier $m$ with accuracy $x + \delta$, we get $q(m) = 0$, $b(m) = x + \delta$ as $p(m) = \frac{x}{x+\delta}$. In that case we can only reduce variance by a factor of $\frac{1}{9}$, or equivalently improve sample efficiency by a factor of $1.125$:

**Proposition 11.** *Fix $q(m) = 0$ and assume $b(m) = x + \delta$ as well as $p(m) = \frac{x}{x+\delta}$ for $x > 0.5$, $\delta > 0$ and $x + \delta \leq 1$. Then*

$$\rho(s(m), \tilde{s}(m))^2 \leq \frac{1}{9}.$$

*Correspondingly, the sample efficiency factor is bounded by $\tau_{\max} \leq 1.125$.*

*Proof.* Fixing $q(m) = 0$, we get

$$\rho(s(m), \tilde{s}(m))^2 = \frac{\left(\frac{x}{x+\delta} - 1\right)(\delta + x - 1)}{1 - \delta}.$$

Taking the $\delta$ derivative yields

$$\frac{d}{d\delta}\rho(s(m), \tilde{s}(m))^2 = -\frac{x(2\delta + x - 1)}{(\delta - 1)^2(\delta + x)^2}.$$

This is zero precisely when $\delta = \frac{1-x}{2}$, and negative for $\delta$ larger than that. Correspondingly, $\delta = \frac{1-x}{2}$ is the global maximum. Inserting this back, we obtain

$$\rho(s(m), \tilde{s}(m))^2 \leq \frac{(x-1)^2}{(x+1)^2}$$

with negative derivative $\frac{d}{dx}\frac{(x-1)^2}{(x+1)^2} = \frac{4(x-1)}{(x+1)^3}$. Correspondingly, the upper bound is maximized at the smallest possible $x = 0.5$, where it equals $\frac{0.5^2}{1.5^2} = \frac{1}{9}$.    □

### A.2    BALANCED AGREEMENT RATE AS AN ALTERNATIVE

Intuitively, high agreement fails to constrain the squared correlation $\rho^2$ because predictors without any signal, such as constant predictors, can achieve high agreement. This is analogous to how accuracy can be misleading for imbalanced classification problems. In classification, this is solved by focusing on balanced accuracy instead. By analogy, we introduce the *balanced agreement* of $\tilde{s}$ with $s$ as

$$\mathrm{BA}(m) := \frac{q(m) + p(m)}{2}.$$

The next theorem confirms that controlling $\mathrm{BA}(m)$ yields meaningful lower bounds on the squared correlation.

**Theorem 12.** *For any value of $\mathrm{BA}(m)$, we have:*

$$4b(m)(1 - b(m))(2\,\mathrm{BA}(m) - 1)^2 \leq \rho(s(m), \tilde{s}(m))^2 \leq |2\,\mathrm{BA}(m) - 1|.$$

Theorem 12 is proven in Appendix C.4. It also provides another upper bound on sample efficiency gains from access to $\tilde{s}$: Whenever $\mathrm{BA}(m)$ is close to half, access to $\tilde{s}$ can not improve sample efficiency by much and the sample efficiency factor $\tau_{\max} = \frac{1}{1-\rho^2}$ is close to one. As a corollary, we show that whenever the proxy has reasonable balanced accuracy, the minimum of the true positive rate $p$ and true negative rate $q$ upper bounds the squared correlation $\rho^2$:

**Corollary 13.** *Whenever $\mathrm{BA}(m) \geq 0.5$, we have*

$$\rho(s(m), \tilde{s}(m))^2 \leq \min\{p(m), q(m)\}.$$

*Proof.* Given $\mathrm{BA}(m) \geq 0.5$, $|2\,\mathrm{BA}(m) - 1|$ is maximized at the maximal value of $\mathrm{BA}(m)$. But with $\min\{p(m), q(m)\} \leq r$, that is at $\frac{r+1}{2}$. Inserting into Theorem 12 yields the result.    □

Figure 5 shows actual value of the maximal sample efficiency and its lower and upper bounds on $\tau_{\max}$ derived from Theorem 12 for a variety of models. In line with our theoretical results, $\tau_{\max}$ lies consistently between the lower and upper bounds. Interestingly, the lower bound appear to be considerably tighter than the upper bound, that is, $\tau_{\max}$ is much closer to the lower bound than the upper bound throughout the experiments.

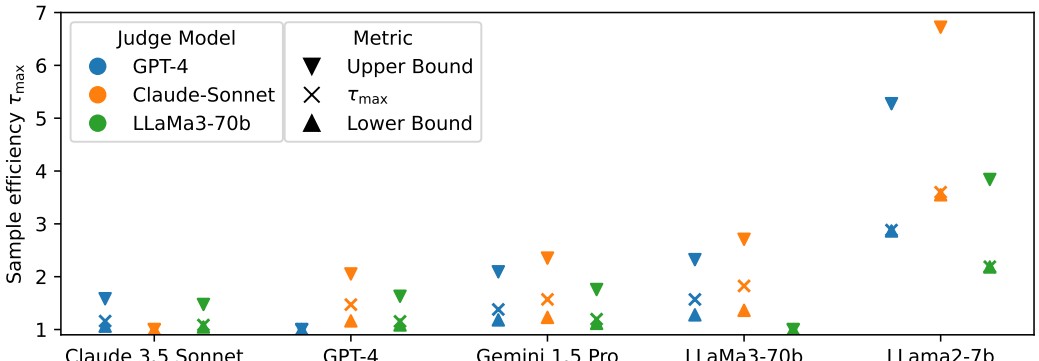

Figure 5: Best possible sample efficiency factor $\tau_{\max}$ and lower/upper bounds based on the balanced accuracy BA for different judges (color) and evaluated models (x-ticks).

### A.3 ESTIMATING THE GAP

If ranking rather than accurately estimating model performance is the main goal, we need to estimate the difference $\mathbb{E}(s(m)) - \mathbb{E}(s(m'))$ for model pairs $(m, m')$. Because of potential cancelations, the difference of the optimal estimators for both terms $\hat{\theta}^{PP}_{\lambda^*(m)}(m) - \hat{\theta}^{PP}_{\lambda^*(m')}(m')$ is not necessarily the optimal estimator for the difference. Correspondingly, further improvements can be made by jointly optimizing $\lambda$ and $\lambda'$ to minimize the variance of $\hat{\theta}^{PP}_{\lambda}(m) - \hat{\theta}^{PP}_{\lambda'}(m')$. Solving the corresponding optimization problem yields:

$$\lambda^*(m) = \frac{\text{Cov}(s(m'), \tilde{s}(m'))\,\text{Cov}(\tilde{s}(m), \tilde{s}(m')) - \text{Cov}(s(m), \tilde{s}(m'))\,\text{Cov}(\tilde{s}(m), \tilde{s}(m'))}{(1 + \frac{n}{N})(\text{Var}(\tilde{s}(m'))\,\text{Var}(\tilde{s}(m)) - \text{Cov}(\tilde{s}(m), \tilde{s}(m')))}$$
$$+ \frac{\text{Cov}(s(m), \tilde{s}(m))\,\text{Var}(\tilde{s}(m')) - \text{Cov}(s(m'), \tilde{s}(m))\,\text{Var}(\tilde{s}(m'))}{(1 + \frac{n}{N})(\text{Var}(\tilde{s}(m'))\,\text{Var}(\tilde{s}(m)) - \text{Cov}(\tilde{s}(m), \tilde{s}(m')))}$$

$$\lambda^*(m') = \frac{\text{Cov}(s(m), \tilde{s}(m))\,\text{Cov}(\tilde{s}(m), \tilde{s}(m')) - \text{Cov}(s(m'), \tilde{s}(m))\,\text{Cov}(\tilde{s}(m), \tilde{s}(m'))}{(1 + \frac{n}{N})(\text{Var}(\tilde{s}(m'))\,\text{Var}(\tilde{s}(m)) - \text{Cov}(\tilde{s}(m), \tilde{s}(m')))}$$
$$+ \frac{\text{Cov}(s(m'), \tilde{s}(m'))\,\text{Var}(\tilde{s}(m)) - \text{Cov}(s(m), \tilde{s}(m'))\,\text{Var}(\tilde{s}(m))}{(1 + \frac{n}{N})(\text{Var}(\tilde{s}(m'))\,\text{Var}(\tilde{s}(m)) - \text{Cov}(\tilde{s}(m), \tilde{s}(m')))}$$

However, using different values of $\lambda$ for different comparisons means that there are multiple different score estimates for model $m$ such that the resulting comparisons might not yield a valid transitive ranking.

## B ADDITIONAL DETAILS ON EXPERIMENTS

For MMLU, we obtain most model predictions from the HELM (Liang et al., 2023) leaderboard and focus on the top 10 models[1]. As HELM does not document model uncertainty, and results for LLama3.1-405B were initially not available, we evaluated LLama3.1-405B in bf16 using 8 A100

---

[1]Cutoff date: 23.07.2024

GPUs and the accelerate library for offloading, ourselves. We use the prompting format from HELM and extract the predicted probabilities $\tilde{p}(y)$ corresponding to the four tokens $y \in Y$ that represent the answer options. For our uncertainty experiment, we renormalize them, setting $p(y) = \frac{\tilde{p}(y)}{\sum_{y' \in Y} \tilde{p}(y')}$.

For MT-bench, we use the results for six models released by the benchmark's authors Zheng et al. (2024). For each triple consisting of two different models $m$, $m'$ and a prompt $x$ there is a varying amount of human expert judgments, as well as a judgment by GPT-4. We aggregate the expert judgments using a majority vote, and discard all triples for which the expert's tied, or GPT-4's judgment amounts to a tie. We then calculate the win-rate for model $m$ by averaging over all remaining triples that include $m$. Note, that this is slightly different from the evaluation in the MT-Bench paper, where win-rates are calculated per model pair, and then averaged over models. MT-Bench also includes a follow-up prompt, with separate judgments for the first and second response. For simplicity, we focus on the first answer.

### B.1 High agreement is not necessary for ranking.

Proposition 2 indicates that a high agreement rate $AG(m)$ is not sufficient for the proxy score $\tilde{s}$ to yield similar model rankings as the real score $s$. However, it turns out that high agreement is also not necessary for stable rankings: For binary classification with judge errors that are independent of the performance of different models, a sufficiently large sample size guarantees correct rankings as long as the error rate is below $50\%$ (Dorner & Hardt, 2024). We demonstrate this phenomenon in Figure 6. In that experiment, the judge reproduces the correct label $60\%$ of the time, but picks a random wrong label otherwise. Despite this judge's low agreement with the correct labels, model rankings are preserved near-perfectly.

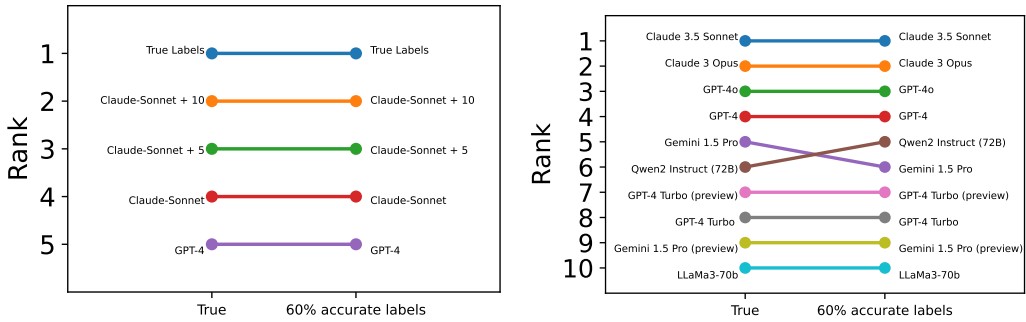

Figure 6: Model rankings on MMLU using a 60% accurate judge with random errors. Ranks are preserved despite agreement substantially below LLM judges.

### B.2 Additional results on TruthfulQA

Figure 7 shows additional results on TruthfulQA. As on MMLU, we extract model answers to the Q&A prompts from HELM[2] (Liang et al., 2023) and define the proxy score $\tilde{s}$ using the judge's answers $m(x)$ as described in Section 2. We use the top three models in the leaderboard as judges, and consider evaluations of these as well as some worse-performing models. As predicted by our theory, the sample efficiency factor $\tau_{\max}$ is consistently below two when weaker models are used to evaluate stronger ones. Beyond that, the sample efficiency factor $\tau_{\max}$ even stays below two when we use the strongest model we considered (Palmyra-x) to judge the weakest (LLaMa2 7B).

### B.3 Additional results on stratification

We consider a debaising method for more sophisticated sampling strategies, namely stratified PPI (Fisch et al., 2024). Note that efficiency improvements from stratification occur independently from

---

[2]We accessed the data on 14.11.2024, but it appears as if the leaderboard has not been updated to include recent models such as GPT-4.

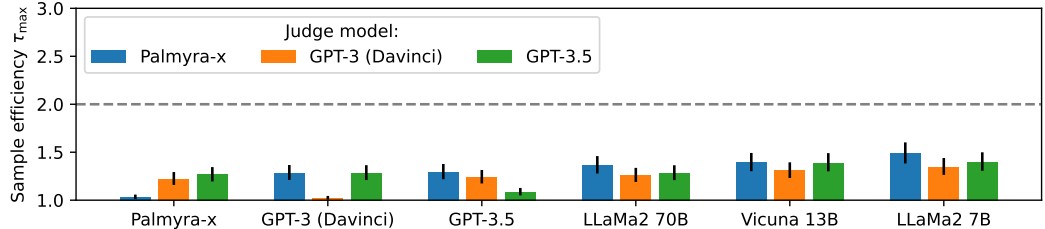

Figure 7: Best possible sample efficiency factor $\tau_{\max}$ using different judges (colors) evaluating different models (x-ticks) on TruthfulQA. Error bars show $90\%$ confidence intervals.

the model-as-judge regime. In order to disentangle gains from stratification from the gains of (stratified) PPI (Fisch et al., 2024), we therefore focus on the *per-stratum* sample efficiency factor $\tau_{\max}$. Theorem 6 still holds *per-stratum* when applying stratified PPI, as long as the assumptions are valid for every stratum. It might however happen that the assumptions do not hold for every stratum, even if they are true globally: A judge model that is weaker than the evaluated model on average could still be stronger on certain strata, thus achieving a sample efficiency factor larger than two (on these strata).

Figure 8 suggests that this might be rare in practice: It shows the per-stratum sample efficiency factors $\tau_{\max}$ on MMLU, using subtasks as strata. Using GPT-4 to judge the stronger Claude (Figure 8a) and LLaMa3.1 405B (Figure 8b), the sample efficiency factors $\tau_{\max}$ varies significantly across strata. They are larger for most strata than the total (non-stratified) sample efficiency factor. However they consistently stay below two. Even when using the stronger LLaMa3.1 405B (Figure 8d) and Claude (Figure 8c) to judge GPT-4, sample efficiency gains above two remain rare: When only observe them in 6 out of 57 (Claude) and 1 out of 57 (LLaMa) cases.

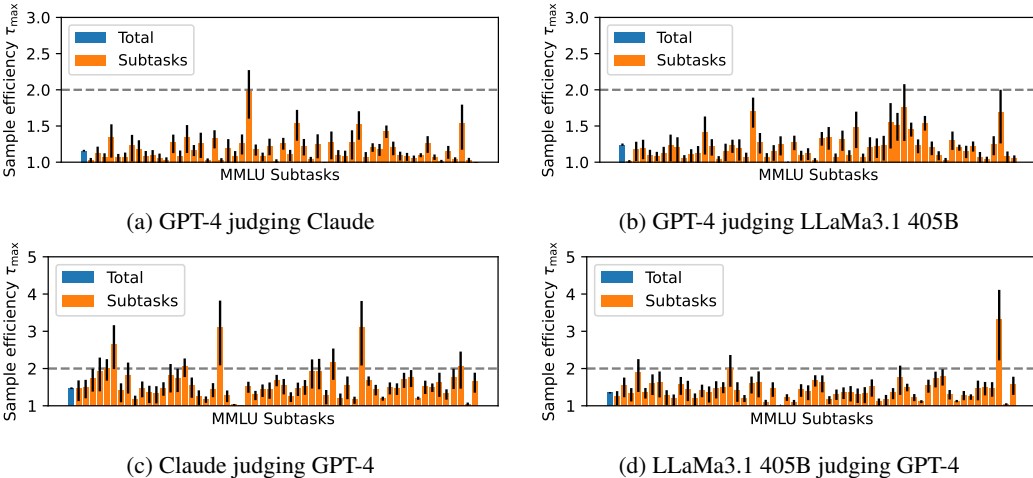

(a) GPT-4 judging Claude

(b) GPT-4 judging LLaMa3.1 405B

(c) Claude judging GPT-4

(d) LLaMa3.1 405B judging GPT-4

Figure 8: Per-stratum sample efficiency factor $\tau_{\max}$ on MMLU. The strata are the 57 MMLU subtasks, in alphabetical order.

## C  PROOFS

### C.1  THEOREM 5

We use a multivariate version of the Cramér-Rao bound, as presented in Lehmann & Casella (2006):

**Theorem 14** (Lehmann & Casella (2006) *Theorem 6.6*). *Consider an experiment with parameter $\theta \in \Omega$ for a product of open sets $\Omega \in \mathbb{R}^n$. Assume the likelihood $l(Z, \theta)$ has the same support for all $\theta$ and finite derivatives $\frac{dl(Z,\theta)}{d\theta}$. Assume the Fisher information $I(\theta) = \mathbb{E}_{Z \sim l(Z,\theta)}\left[\frac{\delta^2 \log l(Z;\theta)}{\delta\theta^2}\right]$*

*is positive definite. Then for any statistic $\delta$, we have $\operatorname{Var}_\theta(\delta) \geq \frac{d\,\mathbb{E}_\theta\,\delta}{d\theta}^t I(\theta)^{-1} \frac{d\,\mathbb{E}_\theta\,\delta}{d\theta}$. In particular, if $\mathbb{E}_\theta\,\delta$ equals the $i$-th component of $\theta$, we have*

$$\operatorname*{Var}_\theta(\delta) \geq (I(\theta)^{-1})_{(i,i)}.$$

We again drop the dependence on $m$ in order to declutter notation. We then use that the likelihoods of independent observations factor and the linearity of derivatives to obtain

$$I(b, q, p) = nI_n(b, q, p) + NI_N(b, q, p),$$

where $I(b, q, p)$ is the Fisher information for our experiment and $I_n(b, q, p)$ and $I_N(b, q, p)$ are the Fisher informations for a single $(s(m), \tilde{s}(m))$ or a single $\tilde{s}(m)$ sample, respectively. Denoting the corresponding likelihoods $l_n$ and $l_N$ respectively, we have:

$$
\begin{aligned}
l_n(s = 0, \tilde{s} = 0) &= (1 - b)q \\
l_n(s = 0, \tilde{s} = 1) &= (1 - b)(1 - q) \\
l_n(s = 1, \tilde{s} = 0) &= b(1 - p) \\
l_n(s = 1, \tilde{s} = 1) &= bp \\
l_N(\tilde{s} = 0) &= (1 - p)(1 - q) + bp \\
l_N(\tilde{s} = 1) &= b(1 - p) + (1 - b)q.
\end{aligned}
$$

Computing the Fisher information[3] yields

$$
\begin{aligned}
(I(\theta))_{(b,b)} =& -\frac{Nb\,(b-1)\,(p+q-1)^2\,(bp - b\,(p-1) - q\,(b-1) + (b-1)\,(q-1))}{b\,(b-1)\,(bp + (b-1)\,(q-1))\,(b\,(p-1) + q\,(b-1))} \\
&- \frac{n\,(bp + (b-1)\,(q-1))\,(b\,(p-1) + q\,(b-1))}{b\,(b-1)\,(bp + (b-1)\,(q-1))\,(b\,(p-1) + q\,(b-1))} \\
(I(\theta))_{(q,q)} =& N\left(\frac{(b-1)^2}{-b\,(p-1) - q\,(b-1)} + \frac{(b-1)^2}{bp + (b-1)\,(q-1)}\right) + n\left(\frac{b-1}{q-1} - \frac{b-1}{q}\right) \\
(I(\theta))_{(p,p)} =& -N\left(\frac{b^2}{b\,(p-1) + q\,(b-1)} - \frac{b^2}{bp + (b-1)\,(q-1)}\right) - n\left(\frac{b}{p-1} - \frac{b}{p}\right) \\
(I(\theta))_{(b,q)} =& \frac{N\,(-bp - bq + b + p + q - 1)}{b^2p^2 + 2b^2pq - 2b^2p + b^2q^2 - 2b^2q + b^2 - 2bpq + bp - 2bq^2 + 3bq - b + q^2 - q} \\
(I(\theta))_{(b,p)} =& \frac{Nb\,(-p - q + 1)}{b^2p^2 + 2b^2pq - 2b^2p + b^2q^2 - 2b^2q + b^2 - 2bpq + bp - 2bq^2 + 3bq - b + q^2 - q} \\
(I(\theta))_{(q,p)} =& \frac{Nb\,(b-1)\,(-bp + b\,(p-1) + q\,(b-1) - (b-1)\,(q-1))}{(bp + (b-1)\,(q-1))\,(b\,(p-1) + q\,(b-1))}.
\end{aligned}
$$

---

[3]We used Sympy for automatic symbolic differentiation and simplification for the remainder of the proof. The code can be found in Proofs.ipynb in the supplementary material.

With this, we can calculate

$$(I(\theta))^{-1}_{(b,b)}$$

$$= \frac{b\left(Nb\left(bp - p + (1-b)(1-q)\right)^2\right)}{n(N+n)(bp + (b-1)(q-1))(bp + (1-b)(1-q) - 1)}$$

$$- \frac{b((N+n)(b-1)(bp + (b-1)(q-1))(bp + (1-b)(1-q) - 1))}{n(N+n)(bp + (b-1)(q-1))(bp + (1-b)(1-q) - 1)}$$

$$= \frac{b\left(-Nb(\mathbb{E}\,\tilde{s} - p))^2 + (1-b)(N+n)\mathbb{E}\,\tilde{s}(1 - \mathbb{E}\,\tilde{s})\right)}{n(N+n)\mathbb{E}\,\tilde{s}(1 - \mathbb{E}\,\tilde{s})}$$

$$= \frac{b(1-b)}{n} - \frac{b\left(Nb(\mathbb{E}\,\tilde{s} - p))^2\right)}{n(N+n)\mathbb{E}\,\tilde{s}(1 - \mathbb{E}\,\tilde{s})}$$

$$= \frac{\mathrm{Var}\,s}{n} - \frac{N}{n(N+n)}\frac{(b\mathbb{E}\,\tilde{s} - bp))^2}{\mathbb{E}\,\tilde{s}(1 - \mathbb{E}\,\tilde{s})}$$

$$= \frac{\mathrm{Var}\,s}{n} - \frac{1}{n + \frac{n^2}{N}}\frac{\mathrm{Cov}(s, \tilde{s})^2}{\mathrm{Var}\,\tilde{s}} = \mathrm{Var}\,\hat{\theta}^{PP}_{\lambda^*}.$$

## C.2 THEOREM 6

*Proof.* To proof the theorem, we reparameterize to simplify the constraints, namely setting $x = \mathrm{AG}(m) \geq 0.5$ and $\delta = b(m) - \mathrm{AG}(m) > 0$. For notational convenience, we also set $q = q(m)$. We then optimize the squared correlation $\rho^2$ over the parameters in succession: At first, we take the derivative with respect to $q$, finding two roots. Thus, the maximum of $\rho^2$ with respect to $q$ is either at one of these roots or the boundary values for $q$. We consider each candidate separately, and optimize over $x$ and $\delta$ in a similar manner. This way, we enumerate all points $q, x, \delta$ that are candidates for the global maximum, i.e. all such points that are feasible, and either hit the boundary of a constraint or have zero derivative respectively, in all of the three variables.

We now delve into the details of the proof: Note that the agreement rate is given by $\mathrm{AG}(m) = (1-b)q + bq$. We therefore replace $p(m) = \frac{(b-1)q + \mathrm{AG}(m)}{b} = \frac{(\delta + x - 1)q + x}{b}$ and obtain a formula for $\rho^2$ based on $q, x$ and $\delta$ :

$$\rho(s(m), \tilde{s}(m))^2 = \frac{(\delta + x - 1)(2\delta q - \delta + 2qx - q)^2}{(\delta + x)(2\delta q - \delta + 2qx - 2q)(2\delta q - \delta + 2qx - 2q + 1)}.$$

*Maximizing in $q \in [0, 1]$* We then take the derivative with respect to $q$ to obtain

$$\frac{d}{dq}\rho(s(m), \tilde{s}(m))^2 = \frac{2(\delta + x - 1)(2\delta q - \delta + 2qx - q)(2\delta qx - \delta q - \delta x + 2qx^2 - 3qx + q)}{(\delta + x)(2\delta q - \delta + 2qx - 2q)^2(2\delta q - \delta + 2qx - 2q + 1)^2}.$$

The numerator is quadratic in $q$, so it has at most two zeros. These turn out to be at $z_1 = \frac{\delta}{2\delta + 2x - 1}$ and $z_2 = \frac{\delta x}{(2x-1)(\delta + x - 1)}$. We note, that there are singularities at $q = \frac{\delta}{2(\delta + x - 1)}$ and $q = \frac{\delta - 1}{2(\delta + x - 1)}$. However, as $\delta + x < 1$, the first singularity occurs at a value of $q \leq 0$. Similarly, the second singularity can be seen to occur at $q \geq 1$, as $\delta - 1 \leq 2\delta + 2x - 2$ is equivalent to $1 \leq \delta + 2x$, which is true as $x \geq 0.5$ and $\delta \geq 0$. Correspondingly, the singularities do not introduce any discontinuities in $q$ for $q \in (0, 1)$. This means that for fixed $x$ and $\delta$, $\rho^2$ is either maximized at $z_1$ or $z_2$, or at the extreme values $q = 0$ or $q = 1$.

**Case 1:** Inserting $q = z_1$ turns the numerator of $\rho^2$ to zero, such that $\rho^2 = 0$.

**Case 2:** We next analyze $z_2 = \frac{\delta x}{(2x-1)(\delta + x - 1)}$: While $\delta x$ and $2x - 1$ are positive by assumption, $\delta + x - 1$ is negative. Correspondingly, $q = z_2$ is not a valid value for $q$ and the maximizer must be one of the other three values.

**Case 3:** Next, inserting $q = 0$ yields

$$\rho(s(m), \tilde{s}(m))^2 = \frac{\delta(\delta + x - 1)}{(\delta - 1)(\delta + x)}.$$

*Maximizing in $x \in [0.5, 1)$* We take the derivative with respect to $x$ to obtain

$$\frac{d}{dx}\rho(s(m), \tilde{s}(m))^2 = \frac{\delta}{(\delta - 1)(\delta + x)^2}.$$

This is negative because $\delta$ is positive while $\delta - 1$ is negative. Correspondingly, it is maximized at the smallest possible value of $x = 0.5$. Inserting $x = 0.5$ yields

$$\rho(s(m), \tilde{s}(m))^2 = \frac{\delta(\delta - 0.5)}{\delta^2 - 0.5\delta - 0.5}.$$

*Maximizing in $\delta \in (0, 1 - x)$* We take the derivative with respect to $\delta$, obtaining

$$\frac{d}{d\delta}\rho(s(m), \tilde{s}(m))^2 = -\frac{(\delta - 0.25)}{(\delta - 1)^2(\delta + 0.5)^2}.$$

This is zero at $\delta = 0.25$ and negative for larger $\delta$, indicating a local maximum. We thus insert $\delta = 0.25$, obtaining

$$\rho(s(m), \tilde{s}(m))^2 = \frac{0.25(0.25 - 0.5)}{0.0625 - 0.125 - 0.5} = \frac{1}{9}.$$

**Case 4:** This leaves us with $q = 1$. We again insert, obtaining

$$\rho(s(m), \tilde{s}(m))^2 = \frac{(\delta + x - 1)(\delta + 2x - 1)}{(\delta + x)(\delta + 2x - 2)}.$$

*Maximizing in $\delta \in (0, 1 - x)$* Taking the derivative with respect to delta yields

$$\frac{d}{d\delta}\rho(s(m), \tilde{s}(m))^2 = \frac{(x - 1)(2\delta + 3x - 2)}{(\delta + x)^2(\delta + 2x - 2)^2}.$$

Note that this only has singularities at $\delta + x = 1$ and $\delta + 2x = 2$, both of which are ruled out by the constraint $\delta + x < 1$. The derivative is zero at $\delta = 1 - \frac{3x}{2}$ and negative for larger $\delta$, indicating a local maximum. When $x > \frac{2}{3}$, this value violates the constraint of $\delta > 0$, such that the maximum has to be at $\delta = 0$ or $\delta = 1 - x$ instead.

*Local maximizing in $x \in (0, \frac{2}{3})$* We insert the local maximizer $\delta = 1 - \frac{3x}{2}$,

$$\rho(s(m), \tilde{s}(m))^2 = \frac{x^2}{(x - 2)^2}$$

with positive derivative

$$\frac{d}{dx}\rho(s(m), \tilde{s}(m))^2 = -\frac{4x}{(x - 2)^3}.$$

This means that the maximum value is at $x = \frac{2}{3}$, where

$$\rho(s(m), \tilde{s}(m))^2 = \frac{1}{4}.$$

*Boundary maximizing in $x \in (0, 1)$* We check the boundaries for $\delta$, i.e., $0 < \delta < 1 - x$. First, we notice that for $\delta \to 1 - x$, $\rho^2 \to 0$. We thus focus on $\delta \to 0$. Inserting yields

$$\rho(s(m), \tilde{s}(m))^2 = \frac{2x - 1}{2x}$$

with positive derivative

$$\frac{d}{dx}\rho(s(m), \tilde{s}(m))^2 = \frac{1}{2x^2}.$$

Correspondingly, it is maximized for $x \to 1$, where it clearly goes to $0.5$. With this, the maximum attainable value for $\rho^2$ equals $0.5$.

$\square$

### C.3 THEOREM 10

Instead of Theorem 10, we prove a stronger version that allows for the judge to be better than the evaluated model. Concretely, the soft error rate of the judge $(1 - \mathrm{SO}(R(\tilde{s})))$ is allowed to be smaller than the soft error rate of the evaluated model $1 - b$ by a factor $\epsilon < 1$. In that case, our upper bound on $\tau$ becomes $\frac{2}{\epsilon}$.

**Theorem 15.** *For any proxy score $\tilde{s}$ with $\epsilon(1 - b) \leq (1 - \mathrm{SO}(R(\tilde{s})))$, we have $\rho^2(s, \tilde{s}) \leq 1 - \frac{\epsilon}{2}$. Correspondingly, the sample effficiency of PPI is bounded: $\tau(\hat{\theta}_{\lambda^*}^{PP}) \leq \frac{2}{\epsilon}$.*

*Proof.* First, we note that the because the Mean squared error/Brier score is a proper scoring rule (Gneiting & Raftery, 2007), the Bayes-optimal predictor $\mathrm{R}(\tilde{s})$ minimizes the mean squared error $\mathrm{MSE}(g(\tilde{s}), s)$ for all post-processing functions $g$. At the same time, we have that the squared correlation $\rho^2(s, \tilde{s}) = R^2(s, f(\tilde{s})) = 1 - \frac{\mathrm{MSE}(s, f(\tilde{s}))}{\mathrm{Var}(s)}$ falls monotonously in $\mathrm{MSE}(s, f(\tilde{s}))$, where $f$ is the affine transformation with the smallest value of $\mathrm{MSE}(s, f(\tilde{s}))$ (Larsen & Marx, 2005). This means that the bayes-optimal predictor $\mathrm{R}(\tilde{s})$ fulfills $\rho^2(s, \mathrm{R}(\tilde{s})) \geq \rho^2(s, \tilde{s})$. As the Bayes-optimal predictor is calibrated, it is sufficient to prove our upper bound for calibrated scores $\tilde{s}$ (i.e. scores $\tilde{s}$ such that $\mathbb{P}(s|\tilde{s} = \gamma) = \gamma$ for all $\gamma \in [0, 1]$.

In the following, we assume $\tilde{s}$ to be calibrated. This means that

$$\mathrm{MSE}(s, \tilde{s}) = \mathbb{E}(s - \tilde{s})^2 = \int_0^1 ((1 - \tilde{s})^2 \tilde{s} + \tilde{s}^2(1 - \tilde{s}))p(\tilde{s})d\tilde{s} = \int_0^1 (1 - \tilde{s})\tilde{s}p(\tilde{s})d\tilde{s},$$

where we use that because of calibration, conditional on a prediction $\tilde{s}$, $s$ is one and the squared error is $(1 - \tilde{s})^2$ with probability $\tilde{s}$. Similarly, the squared error is $\tilde{s}^2$ with probability $(1 - \tilde{s})$.

At the same time, the soft accuracy of $\tilde{s}$ equals

$$\mathrm{SO} = \int_0^1 (\tilde{s}^2 + (1 - \tilde{s})^2)p(\tilde{s})d\tilde{s} = \int_0^1 (2\tilde{s}(\tilde{s} - 1) + 1)p(\tilde{s})d\tilde{s}$$

for calibrated scores $\tilde{s}$, as $s = 1$ yields a score of $\tilde{s}$ and happens with probability $\tilde{s}$, and $s = 0$ happens with probability $(1 - \tilde{s})$ and gets a score of $1 - \tilde{s}$.

By the linearity of the integral, this implies that

$$\mathrm{SO} = 1 - 2\,\mathrm{MSE}$$

or

$$\mathrm{MSE} = \frac{1 - \mathrm{SO}}{2}.$$

As $\tilde{s}$ is bayes-optimal with respect to itself, the optimal affine transformation in terms of $MSE$ is just the identity. Thus, we have that

$$\rho^2(s, \tilde{s}) = R^2(\tilde{s}, s) = 1 - \frac{\mathrm{MSE}}{\mathrm{Var}(s)} = 1 - \frac{\frac{1 - \mathrm{SO}}{2}}{b(1 - b)} = 1 - \frac{1 - \mathrm{SO}}{2b(1 - b)}.$$

This is decreasing in $1 - \mathrm{SO}$ and thus maximized at the minial value of $1 - \mathrm{SO}$, which equals $\epsilon(1 - b)$ by assumption. Inserting yields

$$\rho^2(\tilde{s}, s) \leq 1 - \frac{\epsilon(1 - b)}{2b(1 - b)} = 1 - \frac{\epsilon}{2b}.$$

This is increasing in $b$ and thus maximized at $b = 1$, where we have

$$\rho^2(\tilde{s}, s) \leq 1 - \frac{\epsilon(1 - b)}{2b(1 - b)} = 1 - \frac{\epsilon}{2}.$$

$\square$

Together with a generalized version of Proposition 9, Theorem 15 allows us to generalize theorem 6 from the paper:

**Theorem 16.** *Assume that $b \geq 0.5$. Then for any $0 < \epsilon \leq 1$, and any binary proxy $\tilde{s}$ such that $\mathrm{AG} \geq 0.5$ and $\epsilon(1-b) \leq (1-\mathrm{AG})$, we have $\rho^2 \leq 1 - \frac{\epsilon}{2}$. Correspondingly, the sample efficiency factor $\tau_{\max}$ is at most $\frac{2}{\epsilon}$.*

*Proof.* We first state the generalized version of Proposition 9.

**Proposition 17.** *Assume that $b \geq 0.5$. Then for any $0 < \epsilon \leq 1$, take any binary proxy $\tilde{s}$ such that $\mathrm{AG} \geq 0.5$ and $\epsilon(1-b) \leq (1-\mathrm{AG})$. Then, the recalibrated proxy $R(\tilde{s})$ fulfills $\epsilon(1-b) \leq (1-\mathrm{SO})$.*

We now simply apply Theorem 15. By Proposition 9, the recalibrated version $R(\tilde{s})$ fulfills $\epsilon(1-b) \leq (1-\mathrm{SO})$, yielding the desired bound. In the following, we prove Proposition 17:

*Proof.* We first characterize the recalibrated proxy $R(\tilde{s})$ and its relationship with the real score $s$: $R(\tilde{s})$ equals $\mathbb{P}(s=1|\tilde{s}=1)$ whenever $\tilde{s}=1$ and $\mathbb{P}(s=1|\tilde{s}=0)$ whenever $\tilde{s}=0$. This means that

$$R(\tilde{s}) = \begin{cases} \frac{bp}{bp+(1-b)(1-q)} =: x_1 \ w.p.\ bp + (1-b)(1-q) \\ \frac{b(1-p)}{b(1-p)+(1-b)q} =: x_0 \ w.p.\ b(1-p) + (1-b)q. \end{cases}$$

Plugging this into the definition of soft accuracy yields

$$\mathrm{SO}(R(\tilde{s})) = (bp + (1-b)(1-q))((x_1)^2 + (1-x_1)^2) \\ + (b(1-p) + (1-b)q)((x_0)^2 + (1-x_0)^2)$$

We want to show that $(1-\mathrm{SO}) - \epsilon(1-b) \geq 0$. We begin by inserting the values, obtaining

$$(1-\mathrm{SO}) - \epsilon(1-b) = \frac{(\epsilon(b-1)+1)(bp+(b-1)(q-1))(b(p-1)+q(b-1))}{(bp+(b-1)(q-1))(b(p-1)+q(b-1))} \\ + \frac{(bp+(b-1)(q-1))\left(b^2(p-1)^2 + q^2(b-1)^2\right)}{(bp+(b-1)(q-1))(b(p-1)+q(b-1))} \\ - \frac{(b(p-1)+q(b-1))\left(b^2p^2 + (b-1)^2(q-1)^2\right)}{(bp+(b-1)(q-1))(b(p-1)+q(b-1))}.$$

It is easy to see that the denominator is negative, so it is sufficient to show that the numerator NU is negative. We obtain

$$\mathrm{NU} = (b-1)\left(\epsilon b^2 p^2 + 2\epsilon b^2 pq - 2\epsilon b^2 p + \epsilon b^2 q^2 - 2\epsilon b^2 q + \epsilon b^2 - 2\epsilon bpq + \epsilon bp - 2\epsilon bq^2\right) \\ + (b-1)\left(3\epsilon bq - \epsilon b + \epsilon q^2 - \epsilon q - 2b^2 p^2 + 2b^2 p + 2b^2 q^2 - 2b^2 q - 2bq^2 + 2bq\right).$$

As $b-1$ is negative, it is sufficient to show that $\frac{\mathrm{NU}}{b-1}$ is positive. We take the derivative with respect to $\epsilon$:

$$\frac{d}{d\epsilon}\left(\frac{\mathrm{NU}}{b-1}\right) = (bp + bq - b - q)(bp + bq - b - q + 1).$$

But the first term equals $b(p-1) + (b-1)q = -\mathbb{P}(\tilde{s}=0)$ and is thus between minus one and zero. Correspondingly, the second term is positive, such that the derivative is negative overall. This means that $\left(\frac{\mathrm{NU}}{b-1}\right)$ is minimized at $\epsilon = 1$. We insert that value and take two derivatives with respect to $q$:

$$\frac{d}{dq}\frac{d}{dq}\left(\frac{\mathrm{NU}}{b-1}\right) = 6b^2 - 8b + 2.$$

This has two zeros at $b = \frac{1}{3}$ and $b = 1$ and is negative between them. Correspondingly, under our assumption of $b \geq 0.5$, $\frac{\mathrm{NU}}{b-1}$ is concave in $q$ and thus minimized at either of the two extrema $q = 0$ or $q = 1$. At $q = 0$, we have

$$\frac{\mathrm{NU}}{b-1} = -b(p-1)(bp + b - 1).$$

This is positive whenever
$$bp + b - 1 \geq 0.$$
But at $q = 0$, we have $\mathrm{AG} = bp$ such that the condition becomes
$$\mathrm{AG} + b - 1 \geq 0,$$
which is true because we assumed both $\mathrm{AG}$ and $b$ to be at least $0.5$.

Meanwhile, At $q = 1$, we have
$$\frac{\mathrm{NU}}{b - 1} = -bp\,(bp - 2b + 1).$$
Here it is sufficient to show that
$$bp - 2b + 1 \leq 0.$$
This time, $\mathrm{AG} = bp - b + 1$, such that the condition becomes
$$\mathrm{AG} - b \leq 0,$$
which follows from $\epsilon(1 - b) \leq 1 - \mathrm{AG}$, as $\epsilon$ was fixed to one.

$\square$

$\square$

## C.4 Theorem 12

*Proof.* We again drop the dependence on $m$ for notational convenience.

$$
\begin{aligned}
\rho(s(m), \tilde{s}(m))^2 \\
&= b\frac{(p - ((1 - q)(1 - b) + pb))^2}{((1 - q)(1 - b) + pb)(1 - ((1 - q)(1 - b) + pb))(1 - b)} \\
&= b\frac{((1 - b)(p + q - 1))^2}{(1 - q - b + qb + pb)(1 - (1 - q - b + qb + pb))(1 - b)} \\
&= b(1 - b)\frac{(2\,\mathrm{BA} - 1)^2}{(1 - q - b + 2b\,\mathrm{BA})(q + b - 2b\,\mathrm{BA})}.
\end{aligned}
$$

Clearly, this equals zero when $\mathrm{BA} = 0.5$. Fixing $\mathrm{BA} \neq 0.5$ and taking the derivative with respect to $q$ yields

$$\frac{d}{dq}\rho(s(m), \tilde{s}(m))^2 = -\frac{b(1 - b)(2\,\mathrm{BA} - 1)^2(4b\,\mathrm{BA} - 2b - 2q + 1)}{(1 - q - b + 2b\,\mathrm{BA})^2(q + b - 2b\,\mathrm{BA})^2}.$$

Assuming $0 < b < 1$ and $\mathrm{BA} \neq 0.5$, this is zero if and only if $-(4b\,\mathrm{BA} - 2b - 2q + 1)$ is zero and clearly positive for larger $q$. Correspondingly, as long as the singularities in $\rho^2$ lie outside of the $q$−domain, $q = 2b\,\mathrm{BA} - b + \frac{1}{2}$ is a unique minimum, while $\rho(s(m), \tilde{s}(m))^2$ is maximized either at the minimal or the maximal possible value of $q$.

We first insert the minimum, noting that for a lower bound we do not need to worry about whether this value $q$ is indeed attainable. This yields

$$\rho(s(m), \tilde{s}(m))^2 = 4b(1 - b)(2\,\mathrm{BA} - 1)^2,$$

as both denominator terms reduce to $\frac{1}{2}$.

Next, we assume $\mathrm{BA} > \frac{1}{2}$. As both $p$ and $q$ are in $(0, 1)$, we get that $q = 2\,\mathrm{BA} - p \geq 2\,\mathrm{BA} - 1$. Now, the singularities occur at $q = b(2\,\mathrm{BA} - 1)$ and $q = 2\,\mathrm{BA}\,b - b + 1$, the first of which is smaller than $2\,\mathrm{BA} - 1$, while the second is larger than $1$. Correspondingly, we can ignore them and insert the minimal $q = 2\,\mathrm{BA} - 1$. We obtain

$$\rho(s(m), \tilde{s}(m))^2 = \frac{b(2\,\mathrm{BA} - 1)}{2b\,\mathrm{BA} - 2\,\mathrm{BA} - b + 2},$$

and inserting $q = 1$, we obtain

$$\rho(s(m), \tilde{s}(m))^2 = \frac{(b-1)(2\,\mathrm{BA}-1)}{2b\,\mathrm{BA}-b-1}.$$

These two terms are symmetric in the sense that replacing $b$ in the first term with $1 - b$ yields the second term and vice versa. Thus, it is sufficient to maximize the second term with respect to $b$. We obtain

$$\frac{d}{db}\rho(s(m), \tilde{s}(m))^2 = \frac{(2\,\mathrm{BA}-2)(2\,\mathrm{BA}-1)}{(2b\,\mathrm{BA}-b-1)^2},$$

which is negative such that $\rho(s(m), \tilde{s}(m))^2$ is maximized at $b = 0$, where it equals $2\,\mathrm{BA}-1$.

Similarly, for $\mathrm{BA} < \frac{1}{2}$, $q$ can range from 0 to $2\,\mathrm{BA}$. Again, the singularities are at $q = b(2\,\mathrm{BA}-1)$ and $q = 2\,\mathrm{BA}\,b - b + 1$. This time, the first one is negative, while the second one can easily be seen to be larger than $2\,\mathrm{BA}$. This again allows us to ignore the singularities. Inserting the extreme values yields $\frac{(2\,\mathrm{BA}-1)(b-1)}{2b\,\mathrm{BA}-b+1}$ and $\frac{b(2\,\mathrm{BA}-1)}{2b\,\mathrm{BA}-2\,\mathrm{BA}-b}$, which again are the same after replacing $b$ with $1 - b$. Taking the derivative of the first term with respect to $b$ yields

$$\frac{d}{db}\rho(s(m), \tilde{s}(m))^2 = \frac{(2\,\mathrm{BA})(2\,\mathrm{BA}-1)}{(2b\,\mathrm{BA}-b+1)^2},$$

which is negative as $\mathrm{BA} < \frac{1}{2}$. Correspondingly, $b = 0$ now maximizes $\rho(s(m), \tilde{s}(m))^2$, at which point it equals $1 - 2\,\mathrm{BA}$. Joining both cases, we thus obtain

$$\rho(s(m), \tilde{s}(m))^2 \le |2\,\mathrm{BA}-1|.$$

$\square$

## C.5 PROPOSITION 1

*Proof.*

$$\begin{aligned}
\mathbb{E}\,\tilde{s}(x, m_i) &= \mathbb{P}(\tilde{m}(x) = m_i(x)) \\
&= \mathbb{P}(\tilde{m}(x) = y(x))\,\mathbb{P}(\tilde{m}(x) = m_i(x)|\tilde{m}(x) = y(x)) + \mathbb{P}(m(x) = m_i(x), \tilde{m}(x) \ne y(x)) \\
&= \mathbb{E}\,s(\tilde{m}) \cdot 1 + \mathbb{P}(m_i(x) \ne y(x), \tilde{m}(x) \ne y(x)) \\
&= \mathbb{E}\,s(\tilde{m}) + \mathbb{P}(m_i(x) \ne y(x)) \\
&= \mathbb{E}\,s(\tilde{m}) + 1 - \mathbb{E}\,s(m_j),
\end{aligned}$$

which is monotonously falling in $\mathbb{E}\,s(m_j)$.

$\square$

## C.6 PROPOSITION 2

*Proof.* We again drop the dependence on $m$. In the first case, we need that

$$bp + (1 - b)q = \mathrm{AG} = r = 1 - \mathrm{JB} = 1 - (1 - q)(1 - b) + (1 - p)b.$$

This is equivalent to

$$bp + q - bq = q + b - qb + b - bp$$

or

$$bp = b + b - bp,$$

i.e. $p = 1$. With this, the constraint of $\mathrm{AG} = r$ turns into $b + q(1 - b) = r$. The left side is clearly continuous and monotonous in $q$ and inserting the extreme values of zero and one, we obtain $r = b$ and $r = 1$, such that we can find a solution for all intermediate values of $r$.

In the second case, we want

$$bp + (1 - b)q = \mathrm{AG} = r = \mathrm{JB}+1 = (1 - q)(1 - b) - (1 - p)b + 1.$$

We simplify to get

$$bp + q - bq = 1 - q - b + qb - b + pb + 1$$

or equivalently

$$q - bq = -q - b + qb - b + 2$$

or

$$2q - 2bq + 2b = 2,$$

which is achieved at $q = 1$. In that case, the constraint of $\mathrm{AG} = r$ turns into $bp + (1 - b) = r$ with extreme values again lying at $1 - b = r$ and $1 = r$.

$\square$

### C.7 PROPOSITION 4

*Proof.* We first note that $\operatorname{Var} s(m) = b(m)(1 - b(m))$ and $\operatorname{Var} \tilde{s}(m) = \mathbb{E}\,\tilde{s}(m)(1 - \mathbb{E}\,\tilde{s}(m))$, as both $s$ and $\tilde{s}$ are binary. Furthermore, $\mathbb{E}[s(m)\tilde{s}(m)] = b(m)p(m)$, such that $\operatorname{Cov}(s(m), \tilde{s}(m)) = b(m)(p(m) - \mathbb{E}\,\tilde{s}(m))$. With this, it is easy to see that the formula for $\rho^2$ in the proposition statement is indeed correct.

We then note, that for fixed values of $n$ and $N$, the relative variance reduction of $\hat{\theta}_{\lambda^*}^{PP}$ compared to $\hat{\theta}^{GT}$ is determined by the squared Pearson correlation coefficient $\rho$ between the ground truth labels $s$ and model judgements $\tilde{s}$:

$$\frac{\operatorname{Var} \hat{\theta}_{\lambda^*}^{PP}}{\operatorname{Var} \hat{\theta}^{GT}} = 1 - \frac{1}{1 + \frac{n}{N}}\rho(s(m), \tilde{s}(m))^2.$$

Using the definition of $\tau$, this yields

$$\tau(\hat{\theta}_{\lambda^*}^{PP}) = \frac{1}{1 - \frac{1}{1 + \frac{n}{N}}\rho(s(m), \tilde{s}(m))^2}.$$

Finally, we take the limit of $N \to \infty$ for an upper bound. $\qquad\square$

### C.8 PROPOSITION 8

*Proof.* We drop the parameters' dependence on $m$ and set $q = 1 - b$ and $p = b$. Then JB $= b(1 - b) - (1 - b)b = 0$. At the same time, as there is no judge bias, $\mathbb{E}\,\tilde{s}(m) = b = p$, such that $\rho(s(m), \tilde{s}(m))^2 = 0$. Lastly AG $= b^2 + (1-b)^2$, which can attain any value between $0.5$ and $1$. $\quad\square$

