# OpenReview forum: "Limits to scalable evaluation at the frontier: LLM as judge won’t beat twice the data"
_ICLR.cc/2025/Conference — ICLR 2025 Oral_

### Official Review · Reviewer_PERR · 2024-11-04

**Soundness:** 3
**Presentation:** 3
**Contribution:** 3
**Rating:** 6
**Confidence:** 3

**Summary:**

This paper studies the paradigm of model evaluation in which one uses a capable existing model

**Strengths:**

The paper studies an interesting problem, and presents both theoretical and empirical results supporting its central claim. The theoretical results are non-trivial and actually shed light on the practical question of LLM evaluation. Overall I enjoyed reading the paper.

**Weaknesses:**

The clarity of writing could be somewhat improved, especially the exposition around the experiments, the section feels a bit rushed and it was challenging to relate the content of the figures to the central message of the paper and the theoretical results. I would also suggest increasing the descriptiveness of the figure captions.

The experiments could also be more thorough - currently all experiments are on MMLU, but more datasets would strengthen the empirical results.

See also questions below.

**Questions:**

- I am not very familiar with PPI (S3.3), what is the relation between PPI and doubly robust estimation from causal inference? At least on the surface level \theta^{PP} looks similar to many estimators from causal inference.

- What is the intuition behind the constant factor equalling two exactly?

- Is it straightforward to extend the theory from the binary case to the continuous case (as in the experiments)? Or are there techniques specific to binary outcomes.

---

> ### Author Response · Authors · 2024-11-18
>
> Thank you for the feedback. For the updated pdf, we have rewritten the experiment section to improve clarity and better highlight how the experimental results relate to our theoretical results. We also have added additional details to the figure captions.
>
> Regarding the experiments, we would like to highlight that we also did experiments on MT-Bench, an arena-style benchmark (See Figure 3). As suggested by reviewer eL89, we also added results on TruthfulQA (Appendix B2).
>
> Regarding the questions:
>
> > “Is it straightforward to extend the theory from the binary case to the continuous case (as in the experiments)? Or are there techniques specific to binary outcomes.”
>
> We do not think that an analogon to Theorem 5 (the strict optimality of PPI) is possible for continuous scores, as the likelihoods become non-parametric in that case. As such, we can only hope to obtain results on the sample efficiency gains from PPI rather than for arbitrary debiasing methods. We have added new results on this for the case with a non-binary proxy score to Section 5, making use of a generalized formulation of the agreement score.
>
> > “What is the intuition behind the constant factor equalling two exactly?”
>
> It is difficult to provide a strong intuition for the specific factor of two. The proof of our new Theorem 10 provides some intuition for why the sample efficiency gains are finite: They are equal to the ratio of the Variance of s and the MSE between $\tilde{s}$ and s. Letting both the judge accuracy and the evaluated model’s accuracy go to one has both the MSE and the variance go to zero at a similar rate, yielding a finite factor.
>
> > “What is the relation between PPI and doubly robust estimation from causal inference?”
>
> PPI and doubly robust estimation are closely related. They both aim to debias estimates based on machine learning predictions and often reach identical estimators. Their crucial difference lies in the assumptions made about the machine learning model. Doubly robust estimation, as part of semiparametric inference, typically assumes that the machine learning model estimates the true nuisance function consistently, e.g., at a rate of $o(n^{1/4})$. PPI does not impose these restrictions. The machine learning model is truly treated as an unknown black box. Further discussions can be found in Angelopoulos, et al, 2023.
>
> References:
> Angelopoulos, A. N., Duchi, J. C., & Zrnic, T. (2023). PPI++: Efficient prediction-powered inference. arXiv preprint arXiv:2311.01453.

---

> > ### Comment · Reviewer_PERR · 2024-11-26
> > **Response**
> >
> > Thanks to the authors for their responses and updates to the paper. After reading the other reviews and responses, I'll keep my current score. And thank you for the pointer to PPI++ for the connection to doubly robust estimation.

---

### Official Review · Reviewer_eL89 · 2024-11-04

**Soundness:** 4
**Presentation:** 4
**Contribution:** 4
**Rating:** 8
**Confidence:** 4

**Summary:**

The paper investigates the model-as-judge evaluation framework when a calibration set with ground-truth scores is available to debias the judging LLM. It first shows the gap between prediction accuracy and ranking and then uses the prediction-powered inference as a calibration method to debias the judging LLM. With the debiased prediction, the paper shows that the sample efficiency is upper bounded by 2. The theoretical result is validated in the experiment with several popular LLMs.

**Strengths:**

The theoretical results are solid and the experiment result matches well with the conclusion from the theoretical analysis, which makes the paper very strong.

Novelty and Significance: Giving an upper bound for the sample complexity of LLM-as-a-judge is novel and significant.

Soundness: The theoretical results and the derivation are sound.

**Weaknesses:**

The limitation is discussed in Section 6, which I pretty much agree on. It will be better if weighted PPI such as Stratified PPI can be evaluated in the experiment to show the difference.

Clarity: The clarity can be improved by adding examples of Q&A.

**Questions:**

Will the experiment result still be consistent with Theorem 6 if the tasks are challenging such as TruthfulQA?

---

> ### Author Response · Authors · 2024-11-18
>
> Thank you for the positive review!
>
> We have added additional results on stratified PPI to Appendix B3 in the updated pdf. To disentangle the gains from stratification (which have nothing to do with using an LLM judge) from the gains of (stratified) PPI, we focus on per-stratum sample efficiency gains, using MMLU subtasks as strata. When utilizing a weaker model to evaluate a stronger one, we rarely observe sample efficiency gains of more than a factor two for any stratum. This is perhaps to be expected, as our main theorem still applies for each stratum. So a judge model can only yield sample efficiency gains of more than a factor of 2 on a stratum if it is stronger than the evaluated model on that stratum.
>
> We now provide experiments on TruthfulQA in Appendix B2. We use the top three models from the HELM leaderboard as judges, and find no sample efficiency factor exceeding 1.3. This is substantially below our upper bound of 2, validating our theory.

---

> > ### Comment · Reviewer_eL89 · 2024-11-24
> >
> > Thanks for the response, I keep my rating after reading the revised paper and all other reviews.

---

### Official Review · Reviewer_US4Q · 2024-11-08

**Soundness:** 3
**Presentation:** 2
**Contribution:** 4
**Rating:** 6
**Confidence:** 4

**Summary:**

LLM-as-judge has become a popular practice for assisting humans in labeling and evaluating model performance. While this method shows promise, it still faces critical challenges related to inherent biases. This paper examines the biases inherent in LLM-as-judge setups and reviews existing debiasing methods. Especially, with the rapid advancement of models, there is a growing likelihood that evaluated models may surpass the judge model in performance. The authors demonstrate that when the judge model performs worse than the evaluated model on a given task, even the best debiasing methods offer no advantage over simply doubling the amount of ground truth data.

**Strengths:**

Originality: To the best of my knowledge, this paper is the first to theoretically analyze biases in different judging styles and examine the impact of using state-of-the-art (SOTA) models as judges.

Clarity: The paper covers many important topics, but the structure could benefit from refinement to improve readability. Currently, some sections feel challenging to follow due to a somewhat disjointed flow of ideas.

Significance: With the growing use of LLM-as-judge setups in the research community, this work addresses a highly relevant issue. The paper provides theoretical analysis of inherent biases and explores how sample size impacts the effectiveness of debiasing methods.

**Weaknesses:**

**Structure**: The structure of the paper feels somewhat disjointed, making it difficult to follow the flow of ideas.

**Clarity of Results**: The key takeaways from the final set of experiments (line 452) are not well articulated. The experimental setup and the expected outcomes are unclear.

**Balanced Agreement**: In Section 3.6, where the concept of balance agreement is discussed, but the usage and its validation is missing. I am confused by its usage.

**Questions:**

Line 133: Why is it stated that precisely modeling the specific relationship is infeasible? Could you provide an example to clarify this?

Proposition 1: The explanation is unclear. What does "point-wise" mean in the context of line 166?

In the second setting in the experiment, arena-sylye benchmark, we see sample efficiency lower than 2 when using GPT 4v. but when evaluating on llaam2 -7b it's over 2, what's the examplantion here? is it due to the hugh performance gap between llama2-7b and the judge model?

---

> ### Author Response · Authors · 2024-11-18
>
> Thank you for your thorough feedback.
>
> We took your feedback to heart and made a thorough pass for clarity and readability for the updated version of the pdf. In particular, we have added more detail to the transition between sections to improve the narrative flow and clarify the structure. We are happy to address any further suggestions.
>
> We have added more details on the last experiment on incorporating the uncertainty of judge models and supplemented it with an additional theoretical result. In short, there are two key takeaways: First, making use of a judge model’s uncertainty improves sample efficiency gains. Second, the sample efficiency factor remains clearly below two for evaluating SOTA models. The latter is in line with our new Theorem 10 on sample efficiency gains for non-binary proxies $\tilde{s}$.
>
> Based on your review, we moved the section on balanced agreement to the appendix. We hope this improves the paper’s focus on the main message. Balanced agreement was intended as an intuitive alternative to the commonly reported agreement metric that provides more meaningful bounds on the sample efficiency factor $\tau$. We added a plot for the bounds on $\tau$ afforded by balanced agreement on MMLU to Appendix A, showing that they are valid and meaningful. However, we now believe that directly computing $\tau$ is more useful than computing either agreement or balanced agreement.
>
> Answers to specific questions:
>
> > “Proposition 1: The explanation is unclear. What does "point-wise" mean in the context of line 166?”
>
> By “better in a point-wise sense”, we refer to the condition in Proposition 1 that for all $x \in  X$, $\tilde{m}(x) = y(x)$ implies $m_i(x) = y(x)$”. In other words, a model $\tilde{m}$ is better than another model $m_i$ point-wise if $\tilde{m}$ only makes mistakes on data points $x$, on which $m_i$ is also wrong.
>
> > “Line 133: Why is it stated that precisely modeling the specific relationship is infeasible? Could you provide an example to clarify this?”
>
> Sure. Let us consider MMLU as an example. A model of the specific relationships would be a (parametric) mapping from a question text $x$ and LLM model $m$ to both the correct score $s(x,m)$ (i.e. whether LLM model $m$ correctly answered question $x$) as well as the proxy score $\tilde{s}(x,m)$ (i.e. whether LLM model $m$ correctly answered question $x$ according to the LLM judge). Such a model would be highly complicated. For instance, given the question “Paper will burn at approximately what temperature in Fahrenheit?” from MMLU, it would need to correctly predict both whether GPT-4 answers the question correctly, and whether GPT-4 gives the same answer as Claude.
>
> > “In the second setting in the experiment, arena-sylye benchmark, we see sample efficiency lower than 2 when using GPT 4v. but when evaluating on llaam2 -7b it's over 2, what's the examplantion here? is it due to the hugh performance gap between llama2-7b and the judge model?”
>
> We assume that you are referring to the performance shown in Figure 2. In this case, you are correct. Our theorem states that we can at most get a factor 2 if the evaluated model is better than the judge model. But GPT-4 is a lot better than LLama2-7b, such that the larger sample efficiency factor is not ruled out by our theory. We have added additional details to better explain this in the experiment section.
>
> *We hope that this addressed all your concerns. If so, we would kindly ask you to raise your score to reflect the high relevance and strong contribution of our work indicated in your review.*

---

> > ### Comment · Reviewer_US4Q · 2024-11-25
> >
> > Dear authors,
> >
> > Thanks for your detailed responses. I have visited the latest draft and these transitions and takeaways are great! Thanks for adding Section 4. It looks promising. I would like to highlight the contribution of this paper and raise my scores accordingly. Thanks for your time.

---

### Official Review · Reviewer_hSz6 · 2024-11-09

**Soundness:** 3
**Presentation:** 2
**Contribution:** 3
**Rating:** 6
**Confidence:** 2

**Summary:**

This paper addresses an important challenge in LLM evaluation: the reliance on costly, high-quality annotations, which has become a key bottleneck as model complexity grows. The authors investigate the potential of using large, pre-trained models as "judges" to evaluate newer models, thereby reducing the need for extensive human annotation. They present a theoretical analysis showing a significant limitation: when the judge model’s accuracy is not superior to that of the evaluated model, debiasing methods cannot reduce the necessary amount of ground-truth labels by more than half. Empirical results further underscore this limitation, as practical savings on annotation requirements fall even shorter than the theoretical prediction.

**Strengths:**

1. The authors identify and tackle the misalignment between the ranks produced by the ground-truth score and those produced by LLMs.
2. The motivation for the submission is formalised from the judge's bias and agreement rate perspective.
3. The effect of real samples and the limited benefit of proxy samples for sample efficiency are well modelled and explained.

**Weaknesses:**

1. The writing sometimes is not very easy to follow.
2. The submission only focuses on binary evaluation, can it generalised to more complicated cases?

**Questions:**

See weaknesses section

---

> ### Author Response · Authors · 2024-11-18
>
> Thank you for the positive review.
>
> We made extensive edits to improve the writing in the updated pdf, and would be happy to address any further suggestions.
>
> Regarding the binary evaluations, we would first like to point out that most popular benchmarks use binary evaluations. For example, all tasks in the huggingface open LLM leaderboard (https://huggingface.co/docs/leaderboards/open_llm_leaderboard/about) use binary evaluations. That said, neither the need for debiasing methods nor the specific methods we employ require binary evaluation.
>
> On the theoretical side, moving away from binary scores s and/or proxies $\tilde{s}$ yields non-parametric likelihoods, making an analogon to Theorem 5 incredibly hard. Correspondingly, PPI might not be strictly optimal any more, even though the results from Theorem 5 imply that it is optimal in a certain worst-case formulation.
>
> However, we were able to extend Theorem 6 to the case of **non-binary proxy scores**. The results are presented in Section 5 and are based on a novel generalization of the agreement score to non-binary proxies. Our new theorem 10 suggests that **using PPI**, a weaker judge evaluating a stronger model is not going to improve sample efficiency by more than a factor of two, even for non-binary proxy scores $\tilde{s}$.

---

> > ### Author Response · Authors · 2024-11-29
> >
> > Dear reviewer hSz6,
> >
> > Thank you again for your actionable feedback! We hope our rebuttal has addressed your concerns. As the end of the discussion period approaches, could you please let us know in case you have any additional questions or feedback on the updated paper?

---

### Meta-Review · Area_Chair_twrG · 2024-12-19

**Metareview:**

This submission investigates the limits of using an LLM in lieu of a large set of ground truth labels for evaluating the outputs of another LLM. The setting addressed is one that is becoming increasingly common in the literature: one LLM judges the output of another, and then this estimate is debiased by a small set of data with ground truth labels. The main result of the submission tells us that when the judge LLM has worse or similar accuracy to the LLM being evaluated, the best we can expect is a factor of two improvement in the sample efficiency of the evaluation. The experimental corroboration of this analysis suggests that, in practice, the improvements of LLM as a judge are even more modest.

The reviewers were all excited by novelty and significance of the theoretical results and empirical validation. However, there were also concerns that the presentation was lacking in clarity in some places and there were questions relating to how much the analysis applies beyond binary classification. I see that the submission has undergone a substantial revision to improve clarity, and there is a new section discussing tasks other than binary classification.

**Additional Comments On Reviewer Discussion:**

There was minimal discussion for this paper, as the reviews were already quite positive. The authors did a good job of answering the reviewers' questions and updating the manuscript accordingly.

---

### Decision · Program_Chairs · 2025-01-22

Accept (Oral)